# Generalized Independent Noise Condition
# for Estimating Latent Variable Causal Graphs

**Feng Xie**[*,1,2]**, Ruichu Cai**[*,1,3]**, Biwei Huang**[4]**, Clark Glymour**[4]**, Zhifeng Hao**[1,5]**, Kun Zhang**[*,4]

[1] School of Computer Science, Guangdong University of Technology, Guangzhou, China
[2] School of Mathematical Sciences, Peking University, Beijing, China
[3] Pazhou Lab, Guangzhou, China
[4] Department of Philosophy, Carnegie Mellon University, Pittsburgh, USA
[5] School of Mathematics and Big Data, Foshan University, Foshan, China
xiefeng009@gmail.com, cairuichu@gdut.edu.cn, biweih@andrew.cmu.edu
cg09@andrew.cmu.edu, zfhao@gdut.edu.cn, kunz1@cmu.edu

## Abstract

Causal discovery aims to recover causal structures or models underlying the observed data. Despite its success in certain domains, most existing methods focus on causal relations between observed variables, while in many scenarios the observed ones may not be the underlying causal variables (e.g., image pixels), but are generated by latent causal variables or confounders that are causally related. To this end, in this paper, we consider Linear, Non-Gaussian Latent variable Models (LiNGLaMs), in which latent confounders are also causally related, and propose a Generalized Independent Noise (GIN) condition to estimate such latent variable graphs. Specifically, for two observed random vectors $\mathbf{Y}$ and $\mathbf{Z}$, GIN holds if and only if $\omega^\top \mathbf{Y}$ and $\mathbf{Z}$ are statistically independent, where $\omega$ is a parameter vector characterized from the cross-covariance between $\mathbf{Y}$ and $\mathbf{Z}$. From the graphical view, roughly speaking, GIN implies that causally earlier latent common causes of variables in $\mathbf{Y}$ d-separate $\mathbf{Y}$ from $\mathbf{Z}$. Interestingly, we find that the independent noise condition, i.e., if there is no confounder, causes are independent from the error of regressing the effect on the causes, can be seen as a special case of GIN. Moreover, we show that GIN helps locate latent variables and identify their causal structure, including causal directions. We further develop a recursive learning algorithm to achieve these goals. Experimental results on synthetic and real-world data demonstrate the effectiveness of our method.

## 1 Introduction

Identifying causal relationships from observational data, known as causal discovery, has drawn much attention in the fields of empirical science and artificial intelligence [Spirtes et al., 2010, Pearl, 2019]. Most causal discovery approaches focus on the situation without latent variables, such as the PC algorithm [Spirtes and Glymour, 1991], Greedy Equivalence Search (GES) [Chickering, 2002], and methods based on the Linear, Non-Gaussian Acyclic Model (LiNGAM) [Shimizu et al., 2006], the Additive Noise Model (ANM) [Hoyer et al., 2009], and the Post-NonLinear causal model (PNL) [Zhang and Chan, 2006, Zhang and Hyvärinen, 2009]. However, although these methods have been used in a range of fields, they may fail to produce convincing results in cases with latent variables (or more specifically, confounders), because they do not properly take into account the influences from latent variables as well as many other practical issues [Zhang et al., 2018].

---

[*]These authors contributed equally to this work. The work was done while FX was visiting CMU.

Causal discovery with latent variables has attracted much attention. Some approaches attempt to handle the question based on conditional independence constraints, including the FCI algorithm [Spirtes et al., 1995], RFCI [Colombo et al., 2012], and their variants. They focus on estimating the causal relationships between observed variables rather than that between latent variables. However, in real-world scenarios, it may not be the case—there are also causal relationships between latent variables. Later, it was shown that by utilizing vanishing Tetrad conditions [Spearman, 1928] and, more generally, t-separation, one is able to identify latent variables in linear-Gaussian models [Silva et al., 2006, Sullivant et al., 2010]. Furthermore, by leveraging an extended t-separation [Spirtes, 2013], a more reliable and faster algorithm, called FindOneFactorClusters (FOFC), was developed [Kummerfeld and Ramsey, 2016]. However, these methods may not be able to identify causal directions between latent variables, and they require strong constraints that each latent variable should have at least three pure measurement variables.[2] Such limitation is because they only rely on rank constraints on the covariance matrix, but fail to take into account higher-order statistics. To make use of higher-order information, one may apply overcomplete independent component analysis [Hoyer et al., 2008, Shimizu et al., 2009], but it does not consider the causal structure between latent variables and the size of the equivalence class of the identified structure could be very large [Entner and Hoyer, 2010, Tashiro et al., 2014]. Another interesting work by Anandkumar et al. [2013] extracts second-order statistics in identifying latent factors, while using non-Gaussianity when estimating causal relations between latent variables. Zhang et al. [2017] and Huang* et al. [2020] considered a special type of confounders due to distribution shifts.

Recently, a condition about a particular type of independence relationship between any three variables, called Triad condition, was proposed [Cai et al., 2019], together with the LSTC algorithm to discover the structure between latent variables. Nevertheless, this method does not apply to the case where there are multiple latent variables behind two observed variables.

It is well known that one may use the independent noise condition to recover the causal structure from linear non-Gaussian data without latent variables [Shimizu et al., 2011]. Then a question naturally rises: is it possible to solve the latent-variable problem, by introducing non-Gaussianity and a condition similar to the independent noise condition? Interestingly, we find that it can be achieved by testing the independence between $\omega^\top \mathbf{Y}$ and $\mathbf{Z}$, where $\mathbf{Y}$ and $\mathbf{Z}$ are two observed random vectors, and $\omega$ is a parameter vector based on the cross-covariance between $\mathbf{Y}$ and $\mathbf{Z}$. If $\omega^\top \mathbf{Y}$ and $\mathbf{Z}$ are statistically independent, we term this condition Generalized Independent Noise (GIN) condition. We show that the well-known independent noise condition can be seen as a special case of GIN. From the view of graphical models, roughly speaking, if the GIN condition holds, then in the Linear Non-Gaussian Latent variable Model (LiNGLaM), the causally earlier latent common causes of variables in $\mathbf{Y}$ d-separate $\mathbf{Y}$ from $\mathbf{Z}$. By leveraging GIN, we further develop a practical algorithm to identify important information of the LiNGLaM, including where the latent variables are, the number of latent variables behind any two observed variables, and the causal order of the latent variables.

The contributions of this work are three-fold. 1) We define the GIN condition for an ordered pair of variables sets, provide mathematical conditions that are sufficient for it, and show that the independent noise condition can be seen as its special case. 2) We then further establish a connection between the GIN condition and the graphical patterns in the LiNGLaM, including specific d-separation relations. 3) We exploit GIN to estimate the LiNGLaM, which allows causal relationships between latent variables and multiple latent variables behind any two observed variables. Compared to existing work, a uniquely appealing feature of the proposed method is that it is able to identify the causal order of the latent variables and determine the number of latent variables behind any two observed variables.

## 2 Problem Definition

In this paper, we focus on a particular type of linear acyclic latent variable causal models. We use $\mathbf{V} = \mathbf{X} \cup \mathbf{L}$ to denote the total set of variables, where $\mathbf{X}$ denote the set of observed variables, with $\mathbf{X} = \{X_1, X_2, ... X_m\}$, and $\mathbf{L}$ denote the set of latent variables, with $\mathbf{L} = \{L_1, L_2, ... L_n\}$. We assume that any variable in $\mathbf{V}$ satisfy the following generating process: $V_i = \sum_{k(j)<k(i)} b_{ij} V_j + \varepsilon_{V_i}, i = 1, 2, ..., m + n$, where $k(i)$ represents the causal order of variables in a directed acyclic graph, so that no later variable causes any earlier variable, $b_{ij}$ denotes the causal strength from $V_j$ to $V_i$, and $\varepsilon_{V_i}$ are independent and identically distributed noise variables. Without loss of generality, we assume that all variables have a zero mean (otherwise can be centered). The definition of our model is given below.

**Definition 1** (Linear Non-Gaussian Latent Variable Model (LiNGLaM)). *A LiNGLaM, besides linear and acyclic assumptions, has the following assumptions:*

- *A1. [Measurement Assumption] There is no observed variable in $\mathbf{X}$ being an ancestor of any latent variables in $\mathbf{L}$.[3]*
- *A2. [Non-Gaussianity Assumption] The noise terms are non-Gaussian.*
- *A3. [Double-Pure Child Variable Assumption] Each latent variable set $\mathbf{L}^{'}$, in which every latent variable directly causes the same set of observed variables, has at least $2Dim(\mathbf{L}^{'})$ pure measurement variables as children.[4]*
- *A4. [Purity Assumption] There is no direct edge between observed variables.*

The key difference to existing researches considering linear latent models, such as Bollen [1989], Silva et al. [2006], is that we introduce the assumptions A2~A4, allowing us to identify the casual structure over latent variables, including casual directions. Figure 1 shows a simple example that satisfies the LiNGLaM. For *Non-Gaussianity Assumption*, the non-Gaussian distribution are expected to be ubiquitous, due to Cramér Decomposition Theorem [Cramér, 1962], as stated in Spirtes and Zhang [2016]. Notice that the *Double-Pure Child Variable Assumption* is much milder than that in Tetrad-based methods: for latent variable set $\mathbf{L}^{'}$, we only need $2Dim(\mathbf{L}^{'})$ pure observed variables, while Tetrad needs $2Dim(\mathbf{L}^{'}) + 1$ pure observed variables. In Section 6, we will briefly discuss the situation where Assumption A4 is violated.

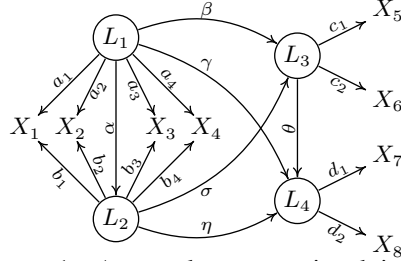

Figure 1: A causal structure involving 4 latent variables and 8 observed variables, where each pair of observed variables in $\{X_1, X_2, X_3, X_4\}$ are affected by two latent variables.

## 3 GIN Condition and Its Implications in LiNGLaM

In this section, we first briefly review the Independent Noise (IN) condition in linear non-Gaussian causal models with no latent variables. Then we formulate the Generalized Independent Noise (GIN) condition and show that it contains the independent noise condition as a special case. We further illustrate how GIN is applied to identify causal relations between latent variables of any two considered groups of observed variables. Finally, we present theoretical results regarding the graphical implications of the GIN condition, which can be used to discover latent variable structures.

### 3.1 Independent Noise Condition in Functional Causal Models

Below, we give the independent noise condition, which has been used in causal discovery of linear, non-Gaussian networks without confounders (e.g., in Shimizu et al. [2011]).

**Definition 2** (IN condition). *Let $Y$ be a single variable and $\mathbf{Z}$ be a set of variables. Suppose all variables follow the linear non-Gaussian acyclic causal model and are observed. We say that $(\mathbf{Z}, Y)$ follows the IN condition, if and only if the residual of regressing $Y$ on $\mathbf{Z}$ is statistically independent from $\mathbf{Z}$. Mathematically, let $\tilde{\omega}$ be the vector of regression coefficients, that is, $\tilde{\omega} := \mathbb{E}[Y\mathbf{Z}^{\top}]\mathbb{E}^{-1}[\mathbf{Z}\mathbf{Z}^{\top}]$; the IN condition holds for $(\mathbf{Z}, Y)$ iff $\tilde{E}_{Y||\mathbf{Z}} = Y - \tilde{\omega}^{\top}\mathbf{Z}$ is independent from $\mathbf{Z}$.*

Lemma 1 in Shimizu et al. [2011] considers the case where $\mathbf{Z}$ is a single variable and shows that $(\mathbf{Z}, Y)$ satisfies the IN condition if and only if $\mathbf{Z}$ is an exogenous (or root) variable relative to $Y$, based on which one can identify the causal relation between $Y$ and $\mathbf{Z}$. As a direct extension of this result, we show that in the case where $\mathbf{Z}$ contains multiple variables, $(\mathbf{Z}, Y)$ satisfies the IN condition if and only if all variables in $\mathbf{Z}$ are causally earlier than $Y$ and there is no common cause behind any variable in $\mathbf{Z}$ and $Y$. This result is given in the following Proposition.

**Proposition 1.** *Suppose all considered variables follow the linear non-Gaussian acyclic causal model and are observed. Let $\mathbf{Z}$ be a subset of those variables and $Y$ be a single variable. Then the following statements are equivalent.*

- *(A) 1) All variables in $\mathbf{Z}$ are causally earlier than $Y$, and 2) there is no common cause for each variable in $\mathbf{Z}$ and $Y$ that is not in $\mathbf{Z}$.*
- *(B) $(\mathbf{Z}, Y)$ satisfies the IN condition.*

## 3.2 Generalized Independent Noise Condition

Below, we first give the definition of the GIN condition, followed by an illustrative example.

**Definition 3** (GIN condition). *Let* $\mathbf{Y}$ *and* $\mathbf{Z}$ *be two observed random vectors. Suppose the variables follow the linear non-Gaussian acyclic causal model. Define the surrogate-variable of* $\mathbf{Y}$ *relative to* $\mathbf{Z}$*, as*

$$E_{\mathbf{Y}||\mathbf{Z}} := \omega^\top \mathbf{Y}, \tag{1}$$

*where* $\omega$ *satisfies* $\omega^\top \mathbb{E}[\mathbf{Y}\mathbf{Z}^\top] = \mathbf{0}$ *and* $\omega \neq \mathbf{0}$*. We say that* $(\mathbf{Z}, \mathbf{Y})$ *follows the GIN condition if and only if* $E_{\mathbf{Y}||\mathbf{Z}}$ *is independent from* $\mathbf{Z}$*.*

In other words, $(\mathbf{Z}, \mathbf{Y})$ violates the GIN condition if and only if $E_{\mathbf{Y}||\mathbf{Z}}$ is dependent on $\mathbf{Z}$. Notice that the Triad condition [Cai et al., 2019] can be seen as a restrictive, special case of the GIN condition, where $\text{Dim}(\mathbf{Y}) = 2$ and $\text{Dim}(\mathbf{Z}) = 1$. We give an example to illustrate that there is a connection between this condition and the causal structure. According to the structure in Figure 1 and by assuming faithfulness, we have that $(\{X_4, X_5\}, \{X_1, X_2, X_3\})$ satisfies the GIN condition, as explained below. The causal models of latent variables is $L_1 = \varepsilon_{L_1}$, $L_2 = \alpha L_1 + \varepsilon_{L_2} = \alpha \varepsilon_{L_1} + \varepsilon_{L_2}$, and $L_3 = \beta L_1 + \sigma L_2 + \varepsilon_{L_3} = (\beta + \alpha\sigma)\varepsilon_{L_1} + \sigma\varepsilon_{L_2} + \varepsilon_{L_3}$, and $\{X_1, X_2, X_3\}$ and $\{X_4, X_5\}$ can then be represented as

$$\underbrace{\begin{bmatrix} X_1 \\ X_2 \\ X_3 \end{bmatrix}}_{\mathbf{Y}} = \begin{bmatrix} a_1 & b_1 \\ a_2 & b_2 \\ a_3 & b_3 \end{bmatrix}\begin{bmatrix} L_1 \\ L_2 \end{bmatrix} + \underbrace{\begin{bmatrix} \varepsilon_{X_1} \\ \varepsilon_{X_2} \\ \varepsilon_{X_3} \end{bmatrix}}_{\mathbf{E_Y}}, \qquad \underbrace{\begin{bmatrix} X_4 \\ X_5 \end{bmatrix}}_{\mathbf{Z}} = \begin{bmatrix} a_4 & b_4 \\ \beta c_1 & \sigma c_1 \end{bmatrix}\begin{bmatrix} L_1 \\ L_2 \end{bmatrix} + \underbrace{\begin{bmatrix} \varepsilon_{X_4} \\ \varepsilon_{X_5'} \end{bmatrix}}_{\mathbf{E_Z}}, \tag{2}$$

where $\varepsilon_{X_5'} = c_2 \varepsilon_{L_3} + \varepsilon_{X_5}$. According to the above equations, $\omega^\top \mathbb{E}[\mathbf{Y}\mathbf{Z}^\top] = \mathbf{0} \Rightarrow \omega = [a_2 b_3 - b_2 a_3, b_1 a_3 - a_1 b_3, a_1 b_2 - b_1 a_2]^\top$. Then we can see $E_{\mathbf{Y}||\mathbf{Z}} = \omega^\top \mathbf{Y} = \omega^\top \mathbf{E_Y}$, and further because $\mathbf{E_Y} \perp\!\!\!\perp \mathbf{Z}$, we have $E_{\mathbf{Y}||\mathbf{Z}} \perp\!\!\!\perp \mathbf{Z}$. That is to say, $(\{X_4, X_5\}, \{X_1, X_2, X_3\})$ satisfies the GIN condition. Intuitively, we have $E_{\mathbf{Y}||\mathbf{Z}} \perp\!\!\!\perp \mathbf{Z}$ because although $\{X_1, X_2, X_3\}$ were generated by $\{L_1, L_2\}$, which are not measurable, $E_{\mathbf{Y}||\mathbf{Z}}$, as a particular linear combination of $\mathbf{Y} = \{X_1, X_2, X_3\}$, successfully removes the influences of $\{L_1, L_2\}$ by properly making use of $\mathbf{Z} = \{X_4, X_5\}$ as a "surrogate".

Next, we discuss a situation where GIN is violated. For example, in this structure, $(\{X_3, X_6\}, \{X_1, X_2, X_5\})$ violates GIN. Specifically, the corresponding variables satisfy the following equations:

$$\underbrace{\begin{bmatrix} X_1 \\ X_2 \\ X_5 \end{bmatrix}}_{\mathbf{Y}} = \begin{bmatrix} a_1 & b_1 \\ a_2 & b_2 \\ \beta c_1 & \sigma c_2 \end{bmatrix}\begin{bmatrix} L_1 \\ L_2 \end{bmatrix} + \underbrace{\begin{bmatrix} \varepsilon_{X_1} \\ \varepsilon_{X_2} \\ \varepsilon_{X_5'} \end{bmatrix}}_{\mathbf{E_Y}}, \qquad \underbrace{\begin{bmatrix} X_3 \\ X_6 \end{bmatrix}}_{\mathbf{Z}} = \begin{bmatrix} a_3 & b_3 \\ \beta c_2 & \sigma c_2 \end{bmatrix}\begin{bmatrix} L_1 \\ L_2 \end{bmatrix} + \underbrace{\begin{bmatrix} \varepsilon_{X_3} \\ \varepsilon_{X_6'} \end{bmatrix}}_{\mathbf{E_Z}}, \tag{3}$$

where $\varepsilon_{X_6'} = c_2 \varepsilon_{L_3} + \varepsilon_{X_6}$. Then under faithfulness assumption, we can see $\omega^T \mathbf{Y} \not\!\perp\!\!\!\perp \mathbf{Z}$ because $\mathbf{E_Y} \not\!\perp\!\!\!\perp \mathbf{E_Z}$ (there exists common component $\varepsilon_{L_3}$ for $\varepsilon_{X_5'}$ and $\varepsilon_{X_6'}$), no matter $\omega^\top \mathbb{E}[\mathbf{Y}\mathbf{Z}^\top] = \mathbf{0}$ or not.

In Section 3.3, we will further investigate graphical implications of GIN in LiNGLaM. For the example given in Figure 1, we have the following observation. $(\{X_4, X_5\}, \{X_1, X_2, X_3\})$ satisfies the GIN condition, and $\{L_1, L_2\}$, the latent common causes for $\{X_1, X_2, X_3\}$, d-separate $\{X_1, X_2, X_3\}$ from $\{X_4, X_5\}$. In contrast, $(\{X_3, X_6\}, \{X_1, X_2, X_5\})$ violates GIN, and $\{X_1, X_2, X_5\}$ and $\{X_3, X_6\}$ are *not* d-separated conditioning on $\{L_1, L_2\}$, the latent common causes of $\{X_1, X_2, X_5\}$.

The following theorem gives mathematical characterizations of the GIN condition, by providing sufficient conditions for when $(\mathbf{Z}, \mathbf{Y})$ satisfies the GIN condition. In the next subsection, we give its implication in LiNGLaM; thanks to the constraints implied by the LiNGLaM, one is able to provide sufficient graphical conditions for GIN to hold.

**Theorem 1.** *Suppose that random vectors* $\mathbf{L}$*,* $\mathbf{Y}$*, and* $\mathbf{Z}$ *are related in the following way:*

$$\mathbf{Y} = A\mathbf{L} + \mathbf{E}_Y, \tag{4}$$

$$\mathbf{Z} = B\mathbf{L} + \mathbf{E}_Z. \tag{5}$$

*Denote by* $l$ *the dimensionality of* $\mathbf{L}$*. Assume* $A$ *is of full column rank. Then, if 1) $Dim(\mathbf{Y}) > l$, 2)* $\mathbf{E}_Y \perp\!\!\!\perp \mathbf{L}$*, 3)* $\mathbf{E}_Y \perp\!\!\!\perp \mathbf{E}_Z$*,[5] and 4) The cross-covariance matrix of* $\mathbf{L}$ *and* $\mathbf{Z}$*,* $\mathbf{\Sigma}_{LZ} = \mathbb{E}[\mathbf{L}\mathbf{Z}^\top]$ *has rank* $l$*, then* $E_{\mathbf{Y}||\mathbf{Z}} \perp\!\!\!\perp \mathbf{Z}$*, i.e.,* $(\mathbf{Z}, \mathbf{Y})$ *satisfies the GIN condition.*

Theorem 1 gives the mathematical conditions, under which $(\mathbf{Z}, \mathbf{Y})$ satisfies the GIN condition. Continue the example in Figure 1. Let $\mathbf{Z} = \{X_4, X_5\}$ and $\mathbf{Y} = \{X_1, X_2, X_3\}$, and thus $\mathbf{L} = \{L_1, L_2\}$. One then can find the following facts: $\text{Dim}(\mathbf{Y}) = 2 > l$, $\mathbf{E}_Y \perp\!\!\!\perp \mathbf{L}$ and $\mathbf{E}_Y \perp\!\!\!\perp \mathbf{E}_Z$ according to Eq. 2, and $\mathbf{\Sigma}_{LZ} = \mathbb{E}[\mathbf{LZ}^\top]$ has full row rank, i.e., 2. Therefore, $(\mathbf{Z}, \mathbf{Y})$ satisfies the GIN condition. All proofs are given in Supplementary Material.

The following proposition shows that the IN condition can be seen as a special case of the GIN condition with $\mathbf{E}_Z = 0$ (i.e., $\mathbf{Z}$ and $\mathbf{L}$ are linearly deterministically related).

**Proposition 2.** *Let $\ddot{Y} := (Y, \mathbf{Z})$. Then the following statements hold:*

1. *$(\mathbf{Z}, \ddot{Y})$ follows the GIN condition if and only if $(\mathbf{Z}, Y)$ follows it.*
2. *If $(\mathbf{Z}, Y)$ follows the IN condition, then $(\mathbf{Z}, \ddot{Y})$ follows the GIN condition.*

Proposition 2 inspires a unified method to handle causal relations between latent variables and those between latent and observed variables. Please see the discussion in Section 6 for more details.

### 3.3 Graphical Criteria of GIN in Terms of LiNGLaM

In this section, we investigate graphical implications of the GIN condition in LiNGLaM, which then inspires us to exploit the GIN condition to discover the graph containing latent variables. Specifically, first, the following theorem shows the connection between GIN and the graphical properties of the variables in terms of LiNGLaM. We denote by $L(X_q)$ the set of latent variables that are the parents of $X_q$ and by $L(\mathbf{Y})$ the set of latent variables that are parents of any component of $\mathbf{Y}$. We say variable set $\mathcal{S}_1$ is an *exogenous set* relative to variable set $\mathcal{S}_2$ if and only if 1) $\mathcal{S}_2 \subseteq \mathcal{S}_1$ or 2) for any variable $V$ that is in $\mathcal{S}_2$ but not in $\mathcal{S}_1$, according to the causal graph over $\{V\} \cup \mathcal{S}_1$ and the ancestors of variables in $\{V\} \cup \mathcal{S}_1$, $V$ does not cause any variable in $\mathcal{S}_1$, and the common cause for $V$ and each variable in $\mathcal{S}_1$, if there is any, is also in $\mathcal{S}_1$ (i.e., relative to $\{V\} \cup \mathcal{S}_1$, $V$ does not cause and is not confounded with any variable in $\mathcal{S}_1$). For instance, according to the structure in Figure 1, let $\mathcal{S}_1 = \{L_1\}$ and $\mathcal{S}_2 = \{L_3, L_4\}$, $\mathcal{S}_1$ is an exogenous set relative to $\mathcal{S}_2$. In constrast, if $\mathcal{S}_1 = \{L_2, L_3\}$ and $\mathcal{S}_2 = \{L_3, L_4\}$, $\mathcal{S}_1$ is not an exogenous set relative to $\mathcal{S}_2$, because $L_4$, which is in $\mathcal{S}_2$ but not in $\mathcal{S}_1$, and $L_2$ (as well as $L_3$), which is in $\mathcal{S}_1$, has a common cause, $L_1$, that is not in $\mathcal{S}_1$.

**Theorem 2.** *Let $\mathbf{Y}$ and $\mathbf{Z}$ be two disjoint subsets of the observed variables of a LiNGLaM. Assume faithfulness holds for the LiNGLaM. $(\mathbf{Z}, \mathbf{Y})$ satisfies the GIN condition if and only if there exists a $k$-size subset of the latent variables $\mathbf{L}$, $0 \le k \le min(Dim(\mathbf{Y}) - 1, Dim(\mathbf{Z}))$, denoted by $\mathcal{S}_L^k$, such that 1) $\mathcal{S}_L^k$ is an exogenous set relative to $L(\mathbf{Y})$, that 2) $\mathcal{S}_L^k$ d-separates $\mathbf{Y}$ from $\mathbf{Z}$, and that 3) the covariance matrix of $\mathcal{S}_L^k$ and $\mathbf{Z}$ has rank $k$, and so does that of $\mathcal{S}_L^k$ and $\mathbf{Y}$.*

Roughly speaking, $S_1$ is an exogenous set relative to $S_2$ if $S_1$ contains causally earlier variables (according to the causal order) in or before $S_2$. Hence, intuitively, the theorem states that $(\mathbf{Z}, \mathbf{Y})$ satisfies the GIN condition when causally earlier common causes of $\mathbf{Y}$ d-separate $\mathbf{Y}$ from $\mathbf{Z}$. We can then see the asymmetry of this condition for $(\mathbf{Z}, \mathbf{Y})$ relative to $L(\mathbf{Y})$ and $L(\mathbf{Z})$. For instance, assuming faithfulness, according to the structure in Figure 1, $(\{X_1, X_2\}, \{X_3, X_4, X_5\})$ satisfies GIN (with $\mathcal{S}_L^2 = \{L_1, L_2\}$), but $(\{X_1, X_6\}, \{X_3, X_4, X_5\})$ does not.

Next, we discuss how to identify the group of observed variables that share the same set of latent direct causes; we call such a set of observed variables a ***causal cluster***. The following theorem formalizes the property of causal clusters and gives a criterion for finding such causal clusters.

**Theorem 3.** *Let $\mathbf{X}$ be the set of all observed variables in a LiNGLaM and $\mathbf{Y}$ be a proper subset of $\mathbf{X}$. If $(\mathbf{X} \setminus \mathbf{Y}, \mathbf{Y})$ follows the GIN condition and there is no subset $\tilde{\mathbf{Y}} \subseteq \mathbf{Y}$ such that $(\mathbf{X} \setminus \tilde{\mathbf{Y}}, \tilde{\mathbf{Y}})$ follows the GIN condition, then $\mathbf{Y}$ is a causal cluster and $Dim(L(\mathbf{Y})) = Dim(\mathbf{Y}) - 1$.*

Consider the example in Figure 1, for $\{X_5, X_6\}$, one can find $(\{X_1, ..., X_4, X_7, X_8\}, \{X_5, X_6\})$ follows the GIN condition, so $\{X_5, X_6\}$ is a causal cluster and $\text{Dim}(L(\{X_5, X_6\})) = \text{Dim}(\{X_5, X_6\}) - 1 = 1$(i.e., $L_3$). But, for $\{X_1, X_2, X_5\}$, $(\{X_3, X_4, X_6, X_7, X_8\}, \{X_1, X_2, X_5\})$ violates the GIN condition, thus $\{X_1, X_2, X_5\}$ is not a causal cluster.

Furthermore, we discuss how to identify the causal direction between latent variables based on their corresponding children. The following theorem shows the asymmetry between the underlying latent variables in terms of the GIN condition.

**Theorem 4.** *Let $\mathcal{S}_p$ and $\mathcal{S}_q$ be two causal clusters of a LiNGLaM. Assume there is no latent confounder behind $L(\mathcal{S}_p)$ and $L(\mathcal{S}_q)$, and $L(\mathcal{S}_p) \cap L(\mathcal{S}_q) = \varnothing$. Further suppose that*

$\mathcal{S}_p$ contains $2Dim(L(\mathcal{S}_p))$ *number of variables with* $\mathcal{S}_p = \{P_1, P_2, ..., P_{2Dim(L(\mathcal{S}_p))}\}$ *and that* $\mathcal{S}_q$ *contains* $2Dim(L(\mathcal{S}_q))$ *number of variables with* $\mathcal{S}_q = \{Q_1, Q_2, ..., Q_{2Dim(L(\mathcal{S}_q))}\}$. *Then if* $(\{P_{Dim(L(\mathcal{S}_p))+1}, ...P_{2Dim(L(\mathcal{S}_p))}\}, \{P_1, ...., P_{Dim(L(\mathcal{S}_p))}, Q_1, ...Q_{Dim(L(\mathcal{S}_q))}\})$ *follows the GIN condition,* $L(\mathcal{S}_p) \rightarrow L(\mathcal{S}_q)$ *holds.*

Consider the example in Figure 1. For two clusters $\{X_1, X_2, X_3, X_4\}$ and $\{X_5, Y_6\}$, where their sets of latent direct causes do not have confounders, one can find $(\{X_3, X_4\}, \{X_1, X_2, X_5\})$ follows GIN condition, so $\{L_1, L_2\} \rightarrow \{L_3\}$.

# 4 GIN Condition-Based Algorithm for Estimating LiNGLaM

In this section, we leverage the above theoretical results and propose a recursive algorithm to discover the structural information of LiNGLaM. The basic idea of the algorithm is that it first finds all causal clusters from the observed data (Step 1), and then it learns the causal order of the latent variables behind these causal clusters (Step 2). The completeness of the algorithm is shown in sections 4.1 (Theorem 3 and Proposition 3 for step 1) and 4.2 (Proposition 4 for step 2).

## 4.1 Step 1: Finding Causal Clusters

To find causal clusters efficiently, one may start with finding clusters with a single latent variable and merge the overlapping culsters, and then increase the number of allowed latent variables until all variables are put in the clusters. We need to consider two practical issues involved in the algorithm. The first is how to find causal clusters and determine how many latent variables they contain, and the second is what clusters should be merged. Theorem 3 answers the first question. Next, for the merge problem, we find that the overlapping clusters can be directly merged into one cluster. This is because the overlapping clusters have the same latent variable as parents in LiNGLaM. The validity of the merging step is guaranteed by Proposition 3, with the algorithm given in Algorithm 1.

**Proposition 3.** *Let* $\mathcal{S}_1$ *and* $\mathcal{S}_2$ *be two clusters of a LiNGLaM and* $Dim(L(\mathcal{S}_1)) = Dim(L(\mathcal{S}_2))$. *If* $\mathcal{S}_1$ *and* $\mathcal{S}_2$ *are overlapping,* $\mathcal{S}_1$ *and* $\mathcal{S}_2$ *share the same set of latent variables as parents.*

---

**Algorithm 1** Identifying Causal Clusters

**Input:** Data set $\mathbf{X} = \{X_1, ..., X_m\}$
**Output:** Causal cluster set $\mathcal{S}$

1: Initialize $\mathcal{S} = \varnothing$, $Len = 1$, and $\mathbf{P} = \mathbf{X}$;
2: **repeat**
3:    **repeat**
4:       Select a variable subset $\mathcal{P}$ from $\mathbf{P}$ such that $Dim(\mathcal{P}) = Len$;
5:       **if** $E_{\mathcal{P}||(\mathbf{P} \backslash \mathcal{P})} \perp\!\!\!\perp (\mathbf{P} \backslash \mathcal{P})$ holds **then**
6:          $\mathcal{S} = \mathcal{S} \cup \mathcal{P}$;
7:       **end if**
8:    **until** all subsets with length $Len$ in $\mathbf{P}$ have been selected;
9:    Merge all the overlapping sets in $\mathcal{S}$;
10:   $\mathbf{P} \leftarrow \mathbf{P} \backslash \mathcal{S}$, and $Len \leftarrow Len + 1$;
11: **until** $\mathbf{P}$ is empty or $Dim(\mathbf{P}) \leq Len$;
12: **Return:** $\mathcal{S}$

---

To test the independence (line 5 in Algorithm 1) between two sets of variables, we check for the pairwise independence with the Fisher's method [Fisher, 1950] instead of testing for the independence between $E_{\mathbf{Y}||\mathbf{Z}}$ and $\mathbf{Z}$ directly. In particular, denote by $p_k$, with $k = 1, 2, ..., c$, all resulting $p$-values from pairwise independence tests. We compute the test statistic as $-2\sum_{k=1}^{c} \log p_k$, which follows the chi-square distribution with $2c$ degrees of freedom when all the pairs are independent.

**Example 1.** *Consider the example in Figure 1. First, we set* $Len = 1$ *to find the clusters with a single latent variable, i.e., we find* $\{X_5, X_6\}$ *and* $\{X_7, X_8\}$ *based on Theorem 3 (Line 4-9). Then we set* $Len = 2$ *and find the clusters* $\{X_1, X_2, X_3, X_4\}$ *with two latent variables.*

## 4.2 Step 2: Learning the Causal Order of Latent Variables

After identifying all clusters, next, we aim to discover the causal order of the set of latent variables of corresponding causal clusters. As an immediate consequence of Theorem 4, the root latent variable can be identified by checking the GIN condition, as stated in the following lemma.

**Lemma 1.** *Let* $\mathcal{S}_r$ *be a cluster and* $\mathcal{S}_k$, $k \neq r$, *be any other cluster of a LiNGLaM. Suppose that* $\mathcal{S}_r$ *contains* $2Dim(L(\mathcal{S}_r))$ *number of variables with* $\mathcal{S}_r = \{R_1, R_2, ..., R_{2Dim(L(\mathcal{S}_r))}\}$ *and that* $\mathcal{S}_k$ *contains* $2Dim(L(\mathcal{S}_k))$ *number of variables with* $\mathcal{S}_k = \{K_1, K_2, ..., K_{2Dim(L(\mathcal{S}_k))}\}$. *if* $(\{R_{Dim(L(\mathcal{S}_r))+1}, ...R_{2Dim(L(\mathcal{S}_r))}\}, \{R_1, ...., R_{Dim(L(\mathcal{S}_r))}, K_1, ...K_{Dim(L(\mathcal{S}_k))}\})$ *follows the GIN condition, then* $L(\mathcal{S}_r)$ *is a root latent variable set.*

Now, the key issue is how to use this lemma to recursively discover the "root variable"[6] until the causal order of latent variables is fully determined. Interestingly, we find that in every iteration, we only need to add the children (i.e., the corresponding causal cluster) of the root variable set into the testing set, such that the number of testing latent variables increases when testing the GIN condition in the following steps. Recall the example discussed in Figure 1. For $L_3$, we find that $(\{X_6, \boxed{X_3, X_4}\}, \{X_5, X_7, \boxed{X_1, X_2}\})$ satisfies the GIN condition, while for $L_4$, $(\{X_8, \boxed{X_3, X_4}\}, \{X_5, X_7, \boxed{X_1, X_2}\})$ violates the GIN condition,[7] which means that $L_3$ is the "root variable". Intuitively speaking, adding the children of the root variable includes the information of the root variable set and create a new "root variable", which helps further remove the effect from them. Accordingly, we have the following proposition to guarantee the correctness of the above process. The details of the process are given in Algorithm 2.

**Proposition 4.** *Suppose that $\{\mathcal{S}_1, ...\mathcal{S}_i, ..., \mathcal{S}_n\}$ contains all clusters of the LiNGLaM. Denote $\mathbf{T} = \{L(\mathcal{S}_1), ...L(\mathcal{S}_i)\}$ and $\mathbf{R} = \{L(\mathcal{S}_{i+1}), ...L(\mathcal{S}_n)\}$, where all elements in $\mathbf{T}$ are causally earlier than those in $\mathbf{R}$. Let $\hat{\mathbf{Z}}$ contain the elements from the half set of the children of each latent variable set in $\mathbf{T}$, and $\hat{\mathbf{Y}}$ contain the elements from the other half set of the children of each latent variable set in $\mathbf{T}$. Furthermore, Let $L(\mathcal{S}_r)$ be a latent variable set of $\mathbf{R}$ and $\mathcal{S}_r = \{R_1, R_2, ..., R_{2Dim(L(\mathcal{S}_r))}\}$. If for any one of the remaining elements $L(\mathcal{S}_k) \in \mathbf{R}$, with $k \neq r$ and $\mathcal{S}_k = \{K_1, K_2, ..., K_{2Dim(L(\mathcal{S}_k))}\}$ such that $(\{R_{Dim(L(\mathcal{S}_r))+1}, ...R_{2Dim(L(\mathcal{S}_r))}, \hat{\mathbf{Z}}\}, \{R_1, ...., R_{Dim(L(\mathcal{S}_r))}, K_1, ...K_{Dim(L(\mathcal{S}_k))}, \hat{\mathbf{Y}}\})$ follows the GIN condition, then $L(\mathcal{S}_r)$ is a root latent variable set in $\mathbf{R}$.*

---

**Algorithm 2** Learning the Causal Order of Latent Variables

**Input:** Set of causal clusters $\mathcal{S}$
**Output:** Causal order $\mathcal{K}$

1: Initialize $\mathcal{L}$ with the root variable sets of each cluster, $\mathbf{T} = \varnothing$, and $\mathcal{K} = \varnothing$;
2: **while** $\mathcal{L} \neq \varnothing$ **do**
3:     Find the root node $L(\mathcal{S}_r)$ according to Proposition 4;
4:     $\mathcal{L} = \mathcal{L} \setminus L(\mathcal{S}_r)$;
5:     Include $L(\mathcal{S}_r)$ into the $\mathcal{K}$;
6:     $\mathbf{T} = \mathbf{T} \cup \mathcal{S}_r$;
7: **end while**
8: **Return:** Causal order $\mathcal{K}$

---

**Example 2.** *Continue to consider the example in Figure 1. We have found the three causal clusters in step 1, i.e., $\mathcal{S}_1 = \{X_1, X_2, X_3\}$, $\mathcal{S}_2 = \{X_5, X_6\}$, and $\mathcal{S}_3 = \{X_7, X_8\}$. Now, we first find that $L(\mathcal{S}_1)$ is the root variable because $(\{X_3, X_4\}, \{X_1, X_2, X_5\})$ and $(\{X_3, X_4\}, \{X_1, X_2, X_7\})$ both satisfy the GIN condition (Line 3). Next, we find $L(\mathcal{S}_2)$ is the "root variable" because $(\{X_6, X_3, X_4\}, \{X5, X7, X_1, X_2\})$ satisfies the GIN condition (Line 3-6). Finally, we return the causal order $\mathcal{K} : L(\mathcal{S}_1) \succ L(\mathcal{S}_2) \succ L(\mathcal{S}_3)$.*

## 5 Experimental Results

To show the efficacy of the proposed approach, we applied it to both synthetic and real-world data. Our source code is available from `https://github.com/xiefeng009/GIN-Condition-for-Estimating-Latent-Variable-Causal-Graphs`.

### 5.1 Synthetic Data

In the following simulation studies, we consider four typical cases: Case 1 & Case 2 have two latent variables $L_1$ and $L_2$, with $L_1 \rightarrow L_2$; Case 3 has three latent variables $L_1$, $L_2$, and $L_3$, with $L_2 \leftarrow L_1 \rightarrow L_3$, and $L_2 \rightarrow L_3$; Case 4 has four latent variables $\{L_1, L_2\}$, $L_3$, and $L_4$, with $\{L_1, L_2\} \rightarrow L_3$, $\{L_1, L_2\} \rightarrow L_4$, and $L_3 \rightarrow L_4$. In all four cases, the data are generated by LiNGLaM and the causal strength $b$ is sampled from a uniform distribution between $[-2, -0.5] \cup [0.5, 2]$, noise terms are generated from uniform[-1,1] variables to the fifth power, and the sample size $N = 500, 1000, 2000$. The details of the graph structures are as follows. [Case 1]: Both $L_1$ and $L_2$ have two pure observed variables, i.e., $L_1 \rightarrow \{X_1, X_2\}$ and $L_2 \rightarrow \{X_3, X_4\}$. [Case 2]: Add extra edges to the graph in Case 1, such that there exist multiple latent variables. In particular, we add two new variables $\{X_5, X_6\}$, such that $\{L1, L2\} \rightarrow \{X_5, X_6\}$, and add the edge $L_1 \rightarrow \{X_3, X_4\}$. [Case 3]: Each latent variable has three pure observed variables, i.e., $L_1 \rightarrow \{X_1, X_2, X_3\}$, $L_2 \rightarrow \{X_4, X_5, X_6\}$,

and $L_3 \rightarrow \{X_7, X_8, X_9\}$. [Case 4]: Add extra latent variables and adjust the observed variables in Case 3 such that it becomes the structure in Figure 1.

We compared our algorithm with BPC [Silva et al., 2006], FOFC [Kummerfeld and Ramsey, 2016],[8] and LSTC [Cai et al., 2019]. We measured the estimation accuracy on two tasks: 1) finding the causal clusters, i.e., locating latent variables, and 2) discovering the causal order of latent variables. Note that BPC and FOFC are only applicable to the first task.

To evaluate the accuracy of the estimated causal cluster, we followed the evaluation metrics from Cai et al. [2019]. Specifically, we used *Latent omission*=$\frac{OL}{TL}$, *Latent commission*=$\frac{FL}{TL}$, and *Mismeasurement*=$\frac{MO}{TO}$, where $OL$ is the number of omitted latent variables, $FL$ is the number of falsely detected latent variables, $TL$ is the total number of latent variables in the ground truth graph, $MO$ is the number of falsely observed variables that have at least one incorrectly measured latent, and $TO$ is the number of observed variables in the ground truth graph. To better evaluate the quality of the estimated causal order, we further used the correct-ordering rate as a metric. Each experiment was repeated 10 times with randomly generated data and the results were averaged. Here, we used the Hilbert-Schmidt Independence Criterion (HSIC) test [Gretton et al., 2008] for the independence test because the data are non-Gaussian.

Table 1: Results by GIN, LSTC, FOFC, and BPC on learning causal clusters.

| Algorithm | | Latent omission | | | | Latent commission | | | | Mismeasurements | | | |
|---|---|---|---|---|---|---|---|---|---|---|---|---|---|
| | | GIN | LSTC | FOFC | BPC | GIN | LSTC | FOFC | BPC | GIN | LSTC | FOFC | BPC |
| *Case 1* | 500 | 0.00(0) | 0.00(0) | 1.00(10) | 0.50(10) | 0.00(0) | 0.00(0) | 0.00(0) | 0.00(0) | 0.00(0) | 0.00(0) | 0.00(0) | 0.00(0) |
| | 1000 | 0.00(0) | 0.00(0) | 1.00(10) | 0.50(10) | 0.00(0) | 0.00(0) | 0.00(0) | 0.00(0) | 0.00(0) | 0.00(0) | 0.00(0) | 0.00(0) |
| | 2000 | 0.00(0) | 0.00(0) | 1.00(10) | 0.50(10) | 0.00(0) | 0.00(0) | 0.00(0) | 0.00(0) | 0.00(0) | 0.00(0) | 0.00(0) | 0.00(0) |
| *Case 2* | 500 | 0.10(2) | 0.20(4) | 0.9(10) | 0.50(10) | 0.00(0) | 0.05(1) | 0.00(0) | 0.00(0) | 0.12(2) | 0.12(4) | 0.00(0) | 0.20(10) |
| | 1000 | 0.05(1) | 0.15(3) | 1.00(10) | 0.50(10) | 0.00(0) | 0.00(0) | 0.00(0) | 0.00(0) | 0.04(1) | 0.12(3) | 0.00(0) | 0.20(10) |
| | 2000 | 0.00(0) | 0.00(0) | 1.00(10) | 0.50(10) | 0.00(0) | 0.02(2) | 0.00(0) | 0.00(0) | 0.00(0) | 0.00(0) | 0.00(0) | 0.20(10) |
| *Case 3* | 500 | 0.20(3) | 0.20(3) | 0.13(9) | 0.10(1) | 0.00(0) | 0.03(3) | 0.00(0) | 0.00(0) | 0.19(3) | 0.17(3) | 0.00(0) | 0.00(0) |
| | 1000 | 0.06(2) | 0.13(2) | 0.16(10) | 0.00(0) | 0.00(0) | 0.00(0) | 0.00(0) | 0.00(0) | 0.06(2) | 0.00(0) | 0.00(0) | 0.00(0) |
| | 2000 | 0.00(0) | 0.00(0) | 0.50(10) | 0.00(0) | 0.00(0) | 0.00(0) | 0.00(0) | 0.00(0) | 0.00(0) | 0.00(0) | 0.00(0) | 0.00(0) |
| *Case 4* | 500 | 0.13(4) | 0.40(6) | 0.90(10) | 0.63(10) | 0.00(0) | 0.23(5) | 0.00(0) | 0.00(0) | 0.04(2) | 0.15(6) | 0.02(2) | 0.06(4) |
| | 1000 | 0.10(3) | 0.26(6) | 0.93(10) | 0.66(10) | 0.00(0) | 0.00(0) | 0.00(0) | 0.00(0) | 0.05(3) | 0.11(2) | 0.01(1) | 0.02(2) |
| | 2000 | 0.03(1) | 0.32(6) | 1.00(10) | 0.70(10) | 0.00(0) | 0.00(0) | 0.00(0) | 0.00(0) | 0.04(1) | 0.11(3) | 0.00(10) | 0.00(0) |

Note: The number in parentheses indicates the number of occurrences that the current algorithm *cannot* correctly solve the problem.

As shown in Table 1, our algorithm, GIN, achieves the best performance (the lowest errors) on almost all cases of the structures. We noticed that although the Mismeasurements of GIN are higher than LSTC in Case 3 when the sample size is small (N=500), the Latent commission of GIN are lower than LSTC. The BPC and FOFC algorithms (with distribution-free tests) do not perform well, which implies that the rank constraints on covariance matrix is not enough to recover more latent structures. Interestingly, although the LSTC algorithm has low errors of the Latent omission in Case 2 (it may be because the structure in Case 2 can be transformed into equivalent pure structures [Cai et al., 2019]), it can not tell us the number of latent variables behind observed variables. Moreover, LSTC fails to recover Case 4 because of the multiple latent variables. The above results demonstrate a clear advantage of our method over the comparisons.

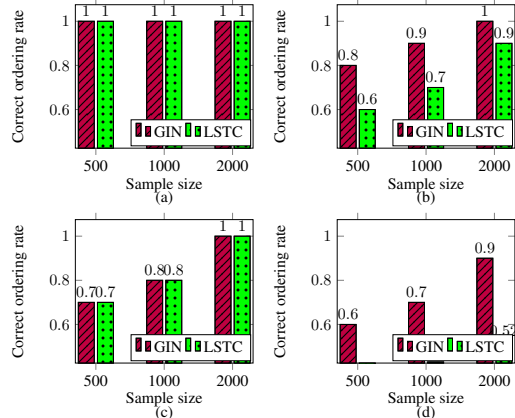

Figure 2: (a-d) Accuracy of the estimated causal order with GIN (purple), and LSTC (green) for Cases 1-4.

Considering that BPC and FOFC algorithms can not discover the causal directions of latent variables, we only reported the results of LSTC algorithm and our algorithm on causal order learning in Figure 2. As shown in Figure 2, the accuracy of the identified causal ordering of our method gradually increases to 1 with the sample size in all the four cases. LSTC can not handle Case 2 & 4. These findings illustrate that our algorithm can discover the correct causal order.

[8]For BPC and FOFC algorithms, we used these implementations in the TETRAD package, which can be downloaded at http://www.phil.cmu.edu/tetrad/.

## 5.2 Real-World Data

Barbara Byrne conducted a study to investigate the impact of organizational (role ambiguity, role conflict, classroom climate, and superior support, etc.) and personality (self-esteem, external locus of control) on three facets of burnout in full-time elementary teachers [Byrne, 2010]. We applied our algorithm to this data set, with 28 observed variables in total.

In the implementation, the kernel width in the HSIC test is set to 0.05. We first applied Algorithm 1 and received six causal clusters,

| Causal Clusters | Observed variables |
|---|---|
| $S_1$ (1) | $RC_1, RC_2, WO_1, WO_2,$ $DM_1, DM_2$ |
| $S_2$ (1) | $CC_1, CC_2, CC_3, CC_4$ |
| $S_3$ (1) | $PS_1, PS_2$ |
| $S_4$ (1) | $ELC_1, ELC_2, ELC_3, ELC_4,$ $ELC_5$ |
| $S_5$ (2) | $SE_1, SE_2, SE_3, EE_1,$ $EE_2, EE_3, DP_1, PA_3$ |
| $S_6$ (3) | $DP_2, PA_1, PA_2$ |

Figure 3: The output of Algorithm 1 in the teacher's burnout study.

including one cluster with 2 latent variables and one cluster with 3 latent variables. The results were given in Table 3. Next, we applied Algorithm 2 and got the final causal order (from root to leaf): $L(S_1) \succ L(S_2) \succ L(S_3) \succ L(S_5) \succ L(S_4) \succ L(S_6)$. Specifically, we had the following findings. 1. The identified clusters are similar to the domain knowledge, e.g., $S_2$ represents the classroom climate, $S_3$ represents the peer support, $S_4$ represents the external locus of control, et al. 2. The learned causal order is similar to Byrne's conclusion, e.g., personal accomplishment ($L(S_6)$) are caused by other latent factors. In addition, role conflict and decision making ($L(S_1)$), classroom climate ($L(S_2)$), and peer support ($L(S_3)$) cause burnout (including emotional exhaustion, depersonalization, and personal accomplishment ($L(S_5)$ and $L(S_6)$)).

# 6 Discussion and Further Work

The preceding sections presented how to use GIN conditions to locate the latent variables and identify their causal structure in the LiNGLaM. In this procedure we examine whether the ordered pair of two disjoint subsets of the observed variables satisfies GIN. As shown in Proposition 2, the GIN condition actually contains IN as a special case, in which the two subsets of variables have overlapping variables. For instance, suppose we have only two variables with $X_1 \to X_2$. Then $(X_1, X_2)$ satisties IN, and

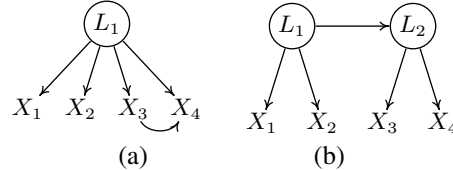

(a)   (b)

Figure 4: Two structures that are distinguishable by the GIN condition, while (a) has an edge between observed variables $X_3$ and $X_4$.

$(X_1, (X_2, X_1))$ satisfies GIN. As a consequence, interestingly, even if we allow edges between observed variables in the LiNGLaM, the GIN condition may also be used to identify them, together with their connections. For instance, in Figure 4(a), $(\{X_1, X_3\}, \{X_2, X_3, X_4\})$ satisfies the GIN condition while $(\{X_1, X_4\}, \{X_2, X_3, X_4\})$ violates the GIN condition, which means that there is an edge between $X_3$ and $X_4$ and $X_3 \to X_4$. In contrast, with the GIN condition on pairs of disjoint subsets of variables, one cannot distinguish between structures (a) and (b). Developing an efficient algorithm that is able to recover the LiNGLaM with directed edges between observed variables in a principled way is part of our future work. Furthermore, in this paper we focus on discovery of the structure of the LiNGLaM, more specifically, the locations of the latent variables and their causal order; as future work, we will also show the (partial) identifiability of the causal coefficients in the model and develop an estimation method for them, to produce a fully specified estimated LiNGLaM (further with edges between observed variables).

# 7 Conclusion

We proposed a Generalized Independent Noise (GIN) condition for estimating a particular type of linear non-Gaussian latent variable causal model, which includes the Independent Noise (IN) condition as a special case. We showed the graphical implications of the GIN condition, based on which we proposed a recursive learning algorithm to locate latent causal variables and identify their causal structure. Experimental results on simulation data and real data further verified the usefulness of our algorithm. Future research along this line includes allowing casual edges between observed variables and allowing nonlinear causal relationships.

**Acknowledgments**

This research was supported in part by Natural Science Foundation of China (61876043) and Science and Technology Planning Project of Guangzhou(201902010058) and Outstanding Young Scientific Research Talents International Cultivation Project Fund of Department of Education of Guangdong Province(40190001). KZ would like to acknowledge the support by the United States Air Force under Contract No. FA8650-17-C-7715. We appreciate the comments from Peter Spirtes and anonymous reviewers, which greatly helped to improve the paper.

## Broader Impact

Causal modeling is a fundamental problem in multiple disciplines of science and data analysis, and causal discovery from observational data has attracted much attention. Existing methods for causal discovery usually assume that there is no confounder (a confounder is a latent direct common cause of two measured variables) or that the confounders for different variables are unrelated. However, it is often the case that observed variables are just reflections of the underlying hidden causal variables, which may be causally related to each other. This is particular true in psychology, neuoscience, and social sciences. Unfortunately, existing methods for finding such latent variables all involve very strong assumptions (e.g., factor analysis assumes that the latent factors are rather low-dimensional and mutually independent), and there is no principle approach to estimating the causal relations between them, especially the causal order. The methodologies and the framework developed in the work have the power to infer the right causal structure, including that over the latent variables, and enable us to correctly understand the systems, which then helps make proper policies, avoid bias or discrimination, and achieve a more transparent and fair world.

## Footnotes

[2]The variable is neither the cause nor the effect of other measurement variables.

[3]Here, this assumption follows the definition in Silva et al. [2006] and it is equivalent to say that there is no observed variable in $\mathbf{X}$ being an parent of any latent variables in $\mathbf{L}$.

[4]$2Dim(\mathbf{L}^{'})$ denotes 2 times the dimension of $\mathbf{L}^{'}$.

[5] Note that we do not assume $\mathbf{E}_Z \perp\!\!\!\perp \mathbf{L}$.

[6]Note that here we call $L$ a "root variable" after we have known the variables that causally earlier than $L$.

[7]Here, the boxes indicate the elements of the root variable set $\{L_1, L_2\}$.

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
