[Supplementary Material]

# Supplement to
# "Generalized Independent Noise Condition
# for Estimating Latent Variable Causal Graphs"

The supplementary material contains

- Proof of Proposition 1;
- Proof of Theorem 1;
- Proof of Proposition 2;
- Proof of (and remark on) Theorem 2;
- Proof of Theorem 3;
- Proof of Theorem 4;
- Proof of Proposition 3;
- Proof of Lemma 1;
- Proof of Proposition 4;
- More experimental results of Synthetic data;
- More details of Real-Word data.

## A  Proofs and Illustrations

We first give an important theorem, which will be used in the proof.

**Darmois-Skitovitch Theorem  [2]** Define two random variables $X_1$ and $X_2$ as linear combinations of independent random variables $e_i(i = 1, ..., p)$:

$$X_1 = \sum_{i=1}^{p} \alpha_i e_i, \qquad X_2 = \sum_{i=1}^{q} \beta_i e_i. \tag{1}$$

Then, if $X_1$ and $X_2$ are independent, all variables $e_j$ for which $\alpha_j \beta_j \neq 0$ are Gaussian. In other words, if there exists a non-Gaussian $e_j$ for which $\alpha_j \beta_j \neq 0$, $X_1$ and $X_2$ are dependent.

### A.1  Proof of Proposition 1

**Proposition 1.** *Suppose all considered variables follow the linear non-Gaussian acyclic causal model. Let $\mathbf{Z}$ be a subset of those variables and $Y$ be a single variable among them. Then the following statements are equivalent.*

- *(A) 1) All variables in $\mathbf{Z}$ are causally earlier than $Y$, and 2) There is no common cause for each variable in $\mathbf{Z}$ and $Y$ that is not in $\mathbf{Z}$.*

- *(B) $(\mathbf{Z}, Y)$ satisfies the IN condition.*

The proof is straightforward, based on the assumption of linear, non-Gaussian acyclic causal models.

## A.2 Proof of Theorem 1

**Theorem 1.** *Suppose that random vectors* $\mathbf{L}$, $\mathbf{Y}$, *and* $\mathbf{Z}$ *are related in the following way:*

$$\mathbf{Y} = A\mathbf{L} + \mathbf{E}_Y, \tag{2}$$

$$\mathbf{Z} = B\mathbf{L} + \mathbf{E}_Z. \tag{3}$$

*Denote by* $l$ *the dimensionality of* $\mathbf{L}$. *Assume* $A$ *is of full column rank. Then, if 1)* $Dim(\mathbf{Y}) > l$, *2)* $\mathbf{E}_Y \perp\!\!\!\perp \mathbf{L}$, *3)* $\mathbf{E}_Y \perp\!\!\!\perp \mathbf{E}_Z$,[1] *and 4) The cross-covariance matrix of* $\mathbf{L}$ *and* $\mathbf{Z}$, $\Sigma_{LZ} = \mathbb{E}[\mathbf{L}\mathbf{Z}^\top]$ *has rank* $l$, *then* $E_{\mathbf{Y}||\mathbf{Z}} \perp\!\!\!\perp \mathbf{Z}$, *i.e.,* $(\mathbf{Z}, \mathbf{Y})$ *satisfies the GIN condition.*

*Proof.* Without loss of generality, assume that each component of $\mathbf{L}$ has a zero mean, and that both $\mathbf{E}_Y$ and $\mathbf{E}_Z$ are zero-mean. If we can find a non-zero vector $\omega$ such that $\omega^\top A = 0$, then $\omega^\top \mathbf{Y} = \omega^\top A\mathbf{L} + \omega^\top \mathbf{E}_Y = \omega^\top \mathbf{E}_Y$, which will be independent from $\mathbf{Z}$ in light of conditions 2) and 3), i.e., the GIN condition for $\mathbf{Y}$ given $\mathbf{Z}$ holds true.

We now construct the vector $\omega$. If conditions 2) and 3) hold, we have $\mathbb{E}[\mathbf{Y}\mathbf{Z}^\top] = A\Sigma_{LZ}$, which is determined by $(\mathbf{Y}, \mathbf{Z})$. We now show that under conditions 4), for any non-zero vector $\omega$, $\omega^\top A = 0$ if and only if $\omega^\top A\Sigma_{\mathbf{LZ}} = 0$ or equivalently $\omega^\top \mathbb{E}[\mathbf{Y}\mathbf{Z}^\top] = 0$ and that such a vector $\omega$ exists.

Suppose $\omega^\top A = 0$, it is trivial to see $\omega^\top A\Sigma_{LZ} = 0$. Notice that condition 4) implies that $\text{rank}(A\Sigma_{LZ}) \leq l$ because $\text{rank}(A\Sigma_{LZ}) \leq \min(\text{rank}(A), \text{rank}(\Sigma_{LZ}))$ and $\text{rank}(A) = l$. Further according to Sylvester Rank Inequality, we have $\text{rank}(A\Sigma_{LZ}) \geq \text{rank}(A) + \text{rank}(\Sigma_{LZ}) - l = l$. Therefore, $\text{rank}(A\Sigma_{LZ}) = l$. Because of condition 1), there must exists a non-zero vector $\omega$, determined by $(\mathbf{Y}, \mathbf{Z})$, such that $\omega^\top \mathbb{E}[\mathbf{Y}\mathbf{Z}^\top] = \omega^\top A\Sigma_{LZ} = 0$. Moreover, this equality implies $\omega^\top A = 0$ because $\Sigma_{LZ}$ has $l$ rows and has rank $l$. With this $\omega$, we have $E_{\mathbf{Y}||\mathbf{Z}} = \omega^\top \mathbf{E}_Y$ and is independent from $\mathbf{Z}$. Thus the theorem holds. $\qquad\square$

## A.3 Proof of Proposition 2

**Proposition 2.** *Let* $\ddot{Y} := (Y, \mathbf{Z})$. *Then the following statements hold:*
1. $(\mathbf{Z}, \ddot{Y})$ *follows the GIN condition if and only if* $(\mathbf{Z}, Y)$ *follows it.*
2. *If* $(\mathbf{Z}, Y)$ *follows the IN condition, then* $(\mathbf{Z}, \ddot{Y})$ *follows the GIN condition.*

*Proof.* For Statement 1, we first show that $(\mathbf{Z}, \ddot{Y})$ follows the GIN condition implies that $(\mathbf{Z}, Y)$ follows the GIN condition. If $(\mathbf{Z}, \ddot{Y})$ follows the GIN condition, then there must exist a non-zero vector $\ddot{\omega}$ so that $\ddot{\omega}^\top \mathbb{E}[\ddot{Y}\mathbf{Z}^\top] = 0$. This equality implies

$$\ddot{\omega}^\top \mathbb{E}\Big[\begin{bmatrix} Y \\ \mathbf{Z} \end{bmatrix} \mathbf{Z}^\top\Big] = \ddot{\omega}^\top \begin{bmatrix} \mathbb{E}[Y\mathbf{Z}^\top] \\ \mathbb{E}[\mathbf{Z}\mathbf{Z}^\top] \end{bmatrix} = \mathbf{0}. \tag{4}$$

Because $\mathbb{E}[\mathbf{Z}\mathbf{Z}^\top]$ is non-singular, we further have

$$\ddot{\omega}^\top \begin{bmatrix} \mathbb{E}[Y\mathbf{Z}^\top]\mathbb{E}^{-1}[\mathbf{Z}\mathbf{Z}^\top] \\ \mathbf{I} \end{bmatrix} = \mathbf{0}.$$

Let $\omega$ be the first $Dim(Y)$ dimensions of $\ddot{\omega}$. Then we have $\omega^\top \mathbb{E}[Y\mathbf{Z}^\top]\mathbb{E}^{-1}[\mathbf{Z}\mathbf{Z}^\top] = \mathbf{0}$, and thus $\omega^\top \mathbb{E}[Y\mathbf{Z}^\top] = \mathbf{0}$. Furthermore, based on the definition of the GIN condition, we have that $E_{\ddot{Y}||\mathbf{Z}} = \ddot{\omega}^\top \ddot{Y}$ is independent from $\mathbf{Z}$. It is easy to see that $E_{Y||\mathbf{Z}} = \omega^\top Y$ is independent from $\mathbf{Z}$. Thus, $(\mathbf{Z}, Y)$ follows the GIN condition.

Next, we show that $(\mathbf{Z}, Y)$ follows the GIN condition implies that $(\mathbf{Z}, \ddot{Y})$ follows the GIN condition. If $(\mathbf{Z}, Y)$ follows the GIN condition, we have

$$\omega^\top \mathbb{E}[Y\mathbf{Z}^\top] = \mathbf{0} \tag{5}$$

Let $\ddot{\omega} = [\omega^\top, \mathbf{0}^\top]^\top$. We have

$$\ddot{\omega}^\top \mathbb{E}[\ddot{Y}\mathbf{Z}^\top] = [\omega^\top, \mathbf{0}^\top]\mathbb{E}\Big[\begin{bmatrix} Y \\ \mathbf{Z} \end{bmatrix}\mathbf{Z}^\top\Big] = \omega^\top \mathbb{E}[Y\mathbf{Z}^\top] = \mathbf{0}. \tag{6}$$

Furthermore, we have $\ddot{\omega}^\mathsf{T}\ddot{Y} = [\omega^\mathsf{T}, 0^\mathsf{T}]\begin{bmatrix} Y \\ \mathbf{Z} \end{bmatrix} = \omega^\mathsf{T}Y$. Based on the definition of GIN, $E_{Y\|\mathbf{Z}} = \omega^\mathsf{T}Y$ is independent from $\mathbf{Z}$. That is to say, $\ddot{\omega}^\mathsf{T}\mathbf{E}_{\ddot{Y}}$ is independent from $\mathbf{Z}$. Thus, $(\mathbf{Z}, \ddot{Y})$ follows the GIN condition.

For Statement 2, If $(\mathbf{Z}, Y)$ follows the IN condition, we have

$$\tilde{\omega} = \mathbb{E}[Y\mathbf{Z}^\mathsf{T}]\mathbb{E}^{-1}[\mathbf{Z}\mathbf{Z}^\mathsf{T}]. \tag{7}$$

Let $\ddot{\omega} = [1^\mathsf{T}, -\tilde{\omega}^\mathsf{T}]^\mathsf{T}$, we get

$$\ddot{\omega}^\mathsf{T}\mathbb{E}[\ddot{Y}\mathbf{Z}^\mathsf{T}] = [1^\mathsf{T}, -\tilde{\omega}^\mathsf{T}]\mathbb{E}\left[\begin{bmatrix} Y \\ \mathbf{Z} \end{bmatrix}\mathbf{Z}^\mathsf{T}\right] = [1^\mathsf{T}, -\tilde{\omega}^\mathsf{T}]\begin{bmatrix} \mathbb{E}[Y\mathbf{Z}^\mathsf{T}] \\ \mathbb{E}[\mathbf{Z}\mathbf{Z}^\mathsf{T}] \end{bmatrix} = \mathbb{E}[Y\mathbf{Z}^\mathsf{T}] - \tilde{\omega}\mathbb{E}[\mathbf{Z}\mathbf{Z}^\mathsf{T}]. \tag{8}$$

From Equations 7 and 8, we have $\ddot{\omega}^\mathsf{T}\mathbb{E}[\ddot{Y}\mathbf{Z}^\mathsf{T}] = 0$. That is to say, $\ddot{\omega}$ satisfies $\ddot{\omega}^\mathsf{T}\mathbb{E}[\ddot{Y}\mathbf{Z}^\mathsf{T}] = 0$ and that $\ddot{\omega}^\mathsf{T} \neq \mathbf{0}$.

Now, we show that $\ddot{\omega}^\mathsf{T}\ddot{Y}$ is independent from $\mathbf{Z}$. We know that $Y - \tilde{\omega}^\mathsf{T}\mathbf{Z}$ is independent from $\mathbf{Z}$ based on the definition of the IN condition. It is easy to see that $\ddot{\omega}^\mathsf{T}\ddot{Y} = [1^\mathsf{T}, -\tilde{\omega}^\mathsf{T}]\begin{bmatrix} Y \\ \mathbf{Z} \end{bmatrix} = Y - \tilde{\omega}^\mathsf{T}\mathbf{Z}$ is independent from $\mathbf{Z}$. Therefore, $(\mathbf{Z}, \ddot{Y})$ follows the GIN condition.

$\square$

### A.4    Proof of and Remark on Theorem 2

**Theorem 2.** *Let $\mathbf{Y}$ and $\mathbf{Z}$ be two disjoint sets of observed variables of a LiNGLaM. Assume faithfulness holds. for the LiNGLaM. $(\mathbf{Z}, \mathbf{Y})$ satisfies the GIN condition (while with the same $\mathbf{Z}$, no proper subset of $\mathbf{Y}$ does) if and only if there exists a $k$-size subset of the latent variables $\mathbf{L}$, $0 \leq k \leq min(Dim(\mathbf{Y}) - 1, Dim(\mathbf{Z}))$, denoted by $\mathcal{S}_L^k$, such that 1) $\mathcal{S}_L^k$ is an exogenous set relative to $L(\mathbf{Y})$, that 2) $\mathcal{S}_L^k$ d-separates $\mathbf{Y}$ from $\mathbf{Z}$, and that 3) the covariance matrix of $\mathcal{S}_L^k$ and $\mathbf{Z}$ has rank $k$, and so does that of $\mathcal{S}_L^k$ and $\mathbf{Y}$.*

*Proof.* The "if" part: First suppose that there exists such a subset of the latent variables, $\mathcal{S}_L^k$, that satisfies the three conditions. Because of condition 1), i.e., that $\mathcal{S}_L^k$ is an exogenous set relative to $L(\mathbf{Y})$ and because according to the LiNGLaM, each $Y_i$ is a linear function of $L(Y_i)$ plus independent noise, we know that $\mathcal{S}_L^k$ is also an exogenous set relative to $\mathbf{Y}$. Hence, we know that each component of $\mathbf{Y}$ can be written as a linear function of $\mathcal{S}_L^k$ and some independent error (which is independent from $\mathcal{S}_L^k$). By a slight abuse of notation, here we use $\mathcal{S}_L^k$ also to denote the vector of the variables in $\mathcal{S}_L^k$. Then we have

$$\mathbf{Y} = A\mathcal{S}_L^k + \mathbf{E}_Y', \tag{9}$$

where $A$ is an appropriate linear transformation, $\mathbf{E}_Y'$ is independent from $\mathcal{S}_L^k$, but its components are not necessarily independent from each other. In fact, according to the LiNGLaM, each observed or hidden variable is a linear combination of the underlying noise terms $\varepsilon_i$. In equation (9), $\mathcal{S}_L^k$ and $\mathbf{E}_Y'$ are linear combinations of disjoint sets of the noise terms $\varepsilon_i$, implied by the directed acyclic structure over all observed and hidden variables.

Let us then write $\mathbf{Z}$ as linear combinations of the noise terms. We then show that because of condition 2), i.e., that $\mathcal{S}_L^k$ d-separates $\mathbf{Y}$ from $\mathbf{Z}$, if any noise term $\varepsilon_i$ is present in $\mathbf{E}_Y'$, it will not be among the noise terms in the expression of $\mathbf{Z}$. Otherwise, if $Z_j$ also involves $\varepsilon_i$, then the direct effect of $\varepsilon_i$, among all observed or hidden variables, is a common cause of $Z_j$ and some component of $\mathbf{Y}$. This path between $Z_j$ and that component of $\mathbf{Y}$, however, cannot be d-separated by $\mathcal{S}_L^k$ because no component of $\mathcal{S}_L^k$ is on the path, as implied by the fact that when $\mathcal{S}_L^k$ is written as a linear combination of the underlying noise terms, $\varepsilon_i$ is not among them. Consequently, any noise term in $\mathbf{E}_Y'$ will not contribute to $\mathcal{S}_L^k$ or $\mathbf{Z}$. Hence, we can express $\mathbf{Z}$ as

$$\mathbf{Z} = B\mathcal{S}_L^k + \mathbf{E}_Z', \tag{10}$$

where $\mathbf{E}'_Z$, which is determined by $\mathcal{S}_L^k$ and $\mathbf{Z}$, is independent from $\mathbf{E}'_Y$. Further considering condition on the dimensionality of $\mathcal{S}_L^k$ and condition 3), one can see that the assumptions in Theorem 1 are satisfied. Therefore, $(\mathbf{Z}, \mathbf{Y})$ satisfies the GIN condition.

The "only-if" part: Then we suppose $(\mathbf{Z}, \mathbf{Y})$ satisfies GIN (while with the same $\mathbf{Z}$, no proper subset of $\mathbf{Y}$ does). Consider all sets $\mathcal{S}_L^k$ that are exogenous relative to $L(\mathbf{Y})$ with $k$ satisfying the condition in the theorem, and we show that at least one of them satisfies conditions 2) and 3). Otherwise, if 2) is always violated, then there is an open path between some leaf node in $L(\mathbf{Y})$, denoted by $L(Y_k)$, and some component of $\mathbf{Z}$, denoted by $Z_j$, and this open path does not go through any common cause of the variables in $L(\mathbf{Y})$. Then they have some common cause that does not cause any other variable in $L(\mathbf{Y})$. Consequently, there exists at least one noise term, denoted by $\varepsilon_i$, that contributes to both $L(Y_k)$ (and hence $Y_k$) and $Z_j$, but not any other variables in $\mathbf{Y}$. Because of the non-Gaussianity of the noise terms and Darmois-Skitovitch Theorem, if any linear projection of $\mathbf{Y}$, $\omega^\top \mathbf{Y}$ is independent from $\mathbf{Z}$, the linear coefficient for $Y_k$ must be zero. Hence $(\mathbf{Z}, \mathbf{Y} \setminus \{Y_k\})$ satisfies GIN, which contradicts the assumption in the theorem. Therefore, there must exists some $\mathcal{S}_L^k$ such that 2) holds. Next, if 3) is violated, i.e., the rank of the covariance matrix of $\mathcal{S}_L^k$ and $\mathbf{Z}$ is smaller than $k$. Then the condition $\omega^\top \mathbb{E}[\mathbf{Y}\mathbf{Z}^\top] = 0$ does not guarantee that $\omega^\top A = 0$. Under the faithfulness assumptions, we then do not have that $\omega^\top \mathbf{Y}$ is independent from $\mathbf{Z}$. Hence, condition 3) also holds. $\qquad\square$

**Remark.** Roughly speaking, the conditions in this theorem can be interpreted the following way: i.) a causally earlier subset (according to the causal order) of the common causes of $\mathbf{Y}$ d-separate $\mathbf{Y}$ from $\mathbf{Z}$, and ii.) the linear transformation from that subset of the common causes to $\mathbf{Z}$ has full column rank. For instance, for the structure in Figure 1 of the main paper, $(\{X_3, X_4\}, \{X_1, X_2, X_5\})$ satisfies GIN, while $(\{X_3, X_6\}, \{X_1, X_2, X_5\})$ does not–note that the difference is that in the latter case one of the variables in $\mathbf{Z}$, $X_6$, is not d-separated from a component of $\mathbf{X}$, which is $X_5$, given the common causes of $\mathbf{X}$. However, when $X_6$ is replaced by $X_4$ in $\mathbf{Z}$, whose direct cause is causally earlier, the d-separation relationship holds, and so is the GIN condition.

### A.5 Proof of Theorem 3

**Theorem 3.** *Let $\mathbf{X}$ be the set of all observed variables in a LiNGLaM and $\mathbf{Y}$ be a proper subset of $\mathbf{X}$. If $(\mathbf{X} \setminus \mathbf{Y}, \mathbf{Y})$ follows the GIN condition and there is no subset $\tilde{\mathbf{Y}} \subseteq \mathbf{Y}$ such that $(\mathbf{X} \setminus \tilde{\mathbf{Y}}, \tilde{\mathbf{Y}})$ follows the GIN condition, then $\mathbf{Y}$ is a causal cluster and $Dim(L(\mathbf{Y})) = Dim(\mathbf{Y}) - 1$.*

*Proof.* We will prove it by contradiction. Let $\mathbf{Y} = (Y_1^\top, ..., Y_{Dim(\mathbf{Y})}^\top)^\top$. There are two cases to consider.

Case 1). Assume that $\mathbf{Y}$ is not a causal cluster and show that $(\mathbf{X} \setminus \mathbf{Y}, \mathbf{Y})$ violates the GIN condition, leading to the contradiction. Since $\mathbf{Y}$ is not a causal cluster, without loss of generality, $L(\mathbf{Y})$ must contain at least two different parental latent variable sets, denoted by $L_a$ and $L_b$. Now, we show that there is no non-zero vector $\omega$ such that $\omega^\top \mathbf{Y}$ is independent from $\mathbf{X} \setminus \tilde{\mathbf{Y}}$. Because there is no subset $\tilde{\mathbf{Y}} \subseteq \mathbf{Y}$ such that $(\mathbf{X} \setminus \tilde{\mathbf{Y}}, \mathbf{Y})$ follows the GIN condition, the number of elements containing the components of $L_a$ in $\mathbf{Y}$ is smaller than $Dim(L_a) + 1$ and the number of elements containing the components of $L_b$ in $\mathbf{Y}$ is less than $Dim(L_b) + 1$. Thus, we obtain that there is no $\omega \neq 0$ such that $\omega^\top \mathbb{E}[\mathbf{Y}((\mathbf{X} \setminus \mathbf{Y})^\top] = 0$. That is to say, $\omega^\top \mathbf{Y}$ is dependent on $\mathbf{X} \setminus \mathbf{Y}$, i.e., $(\mathbf{X} \setminus \mathbf{Y}, \mathbf{Y})$ violates the GIN condition, which leads to the contradiction.

Case 2). Assume that $\mathbf{Y}$ is a causal cluster but $Dim(L(\mathbf{Y})) \neq Dim(\mathbf{Y}) - 1$. First, we consider the case where $Dim(L(\mathbf{Y})) < Dim(\mathbf{Y}) - 1$. If $Dim(L(\mathbf{Y})) > Dim(\mathbf{Y}) - 1$, we always can find a subset $\tilde{\mathbf{Y}} \subseteq \mathbf{Y}$ and $Dim(\tilde{\mathbf{Y}}) = Dim((L(\mathbf{Y})) + 1$ such that $(\mathbf{X} \setminus \tilde{\mathbf{Y}}, \mathbf{Y})$ follows the GIN condition, leading to the contradiction.

We then consider the case where $Dim(L(\mathbf{Y})) > Dim(\mathbf{Y}) - 1$. Due to the linear assumption, each element in $L(\mathbf{Y})$ contains components $\{\varepsilon_{L_1^Y}, ..., \varepsilon_{L_{Dim(L(\mathbf{Y}))}^Y}\}$. Because $Dim(L(\mathbf{Y})) > Dim(\mathbf{Y}) - 1$, $\omega^\top(Y_1^\top, ...., Y_{Dim(\mathbf{Y})}^\top)^\top$ contains $\varepsilon_{L_i^Y}, i \in \{1, .., Dim(L(\mathbf{Y}))\}$, for any $\omega \neq 0$. According to the Darmois-Skitovitch Theorem, we have $\omega^\top(Y_1^\top, ...., Y_{Dim(\mathbf{Y})}^\top)^\top \not\perp\!\!\!\perp \mathbf{X} \setminus \mathbf{Y}$. That is to say, $(\mathbf{X} \setminus \mathbf{Y}, \mathbf{Y})$ violates the GIN condition, which leads to a contradiction. $\qquad\square$

## A.6  Proof of Theorem 4

**Theorem 4.** *Let $\mathcal{S}_p$ and $\mathcal{S}_q$ be two causal clusters of a LiNGLaM. Assume there is no latent confounder for $L(\mathcal{S}_p)$ and $L(\mathcal{S}_q)$, and $L(\mathcal{S}_p) \cap L(\mathcal{S}_q) = \varnothing$. Further suppose that $\mathcal{S}_p$ contains $2Dim(L(\mathcal{S}_p))$ number of variables with $\mathcal{S}_p = \{P_1, P_2, ..., P_{2Dim(L(\mathcal{S}_p))}\}$ and that $\mathbf{C}_q$ contains $2Dim(L(\mathcal{S}_q))$ number of variables with $\mathcal{S}_q = \{Q_1, Q_2, ..., Q_{2Dim(L(\mathcal{S}_q))}\}$. Then if $(\{P_{Dim(L(\mathcal{S}_p))+1}, ...P_{2Dim(L(\mathcal{S}_p))}\}, \{P_1, ...., P_{Dim(L(\mathcal{S}_p))}, Q_1, ...Q_{Dim(L(\mathcal{S}_q))}\})$ follows the GIN condition, $L(\mathcal{S}_p) \to L(\mathcal{S}_q)$ holds.*

*Proof.* For $L(\mathcal{S}_p)$ and $L(\mathcal{S}_q)$, there are two possible causal relations: $L(\mathcal{S}_p) \to L(\mathcal{S}_q)$ and $L(\mathcal{S}_p) \leftarrow L(\mathcal{S}_q)$. For clarity, let $m = Dim(L(\mathcal{S}_p))$ and $n = Dim(L(\mathcal{S}_p))$. Further, Let $L(\mathcal{S}_p) = \{L_1^p, ..., L_m^p\}$ and $L(\mathcal{S}_q) = \{L_1^q, ..., L_n^q\}$ (note that subscripts denote the causal order).

First, we consider case 1: $L(\mathcal{S}_p) \to L(\mathcal{S}_q)$, by leveraging the result of Theorem 1.

According to the linearity assumption, we have

$$\underbrace{\begin{bmatrix} P_1 \\ \vdots \\ P_m \\ Q_1 \\ \vdots \\ Q_n \end{bmatrix}}_{\mathbf{Y}} = \underbrace{\begin{bmatrix} C_{11} & \cdots & C_{m1} \\ \vdots & \ddots & \vdots \\ C_{m1} & \cdots & C_{mm} \\ D_{11} & \cdots & D_{n1} \\ \vdots & \ddots & \vdots \\ D_{n1} & \cdots & D_{nn} \end{bmatrix}}_{A} \underbrace{\begin{bmatrix} L_1^p \\ \vdots \\ L_m^p \end{bmatrix}}_{\mathbf{L}} + \underbrace{\begin{bmatrix} \varepsilon_{P_1} \\ \vdots \\ \varepsilon_{P_m} \\ \varepsilon_{Q_1}' \\ \vdots \\ \varepsilon_{Q_n}' \end{bmatrix}}_{\mathbf{E}_Y} \tag{11}$$

and

$$\underbrace{\begin{bmatrix} P_{m+1} \\ \vdots \\ P_{2m} \end{bmatrix}}_{\mathbf{Z}} = \underbrace{\begin{bmatrix} B_{11} & \cdots & B_{m1} \\ \vdots & \ddots & \vdots \\ B_{m1} & \cdots & B_{mm} \end{bmatrix}}_{B} \underbrace{\begin{bmatrix} L_1^p \\ \vdots \\ L_m^p \end{bmatrix}}_{\mathbf{L}} + \underbrace{\begin{bmatrix} \varepsilon_{P_{m+1}} \\ \vdots \\ \varepsilon_{P_{2m}} \end{bmatrix}}_{\mathbf{E}_Z}, \tag{12}$$

where $\varepsilon_{Q_i}' = \sum_{k=1}^{n} f_k \varepsilon_{L_k^q} + \varepsilon_{Q_i}$.

Now, we verify conditions 1) ~ 4) in Theorem 1. Based on Equations 11 and 12, we have $Dim(\mathbf{L}) = m$. For condition 1), $Dim(\mathbf{Y}) = m + n > m$. For condition 2), $\mathbf{E}_Y = (\varepsilon_{P_1}, ..., \varepsilon_{P_m}, \varepsilon_{Q_1}', ..., \varepsilon_{Q_n}')^\mathsf{T}$ is independent from $\mathbf{L} = \{L_1^p, ..., L_m^p\}$, due to the fact that there is no common component between $\mathbf{E}_Y$ and $\mathbf{L}$ and that each component is independent of each other. For condition 3), because $\varepsilon_{P_k}, k = 1, ..., 2m$, is independent from $\mathbf{L}$, $\mathbf{E}_Z \perp\!\!\!\perp \mathbf{L}$. For condition 4), $\boldsymbol{\Sigma}_{\mathbf{LZ}} = \mathbb{E}[\mathbf{L}\mathbf{Z}^\mathsf{T}] = \boldsymbol{\Sigma}_L B^\mathsf{T}$. Because $Dim(B) = m$, we obtain that $\boldsymbol{\Sigma}_{\mathbf{LZ}}$ has rank $m$. Therefore, $(\{P_{(m+1)+1}, ...P_{2m}\}, \{P_1, ...., P_m, Q_1, ...Q_n\})$ follows the GIN condition.

Next, we consider case 2: $L(\mathcal{S}_p) \leftarrow L(\mathcal{S}_q)$. According to the definition of the GIN condition, we need to find a vector $\omega \neq 0$ such that $\omega^\mathsf{T} \mathbb{E}[(P_1, ...., P_m, Q_1, ...Q_n)(P_{m+1}, ...., P_{2m})^\mathsf{T}] = 0$. Due to the linearity assumption, each element in $\{P_1, ...., P_{2m}\}$ contains the component in $\varepsilon_{L_1^P}, ..., \varepsilon_{L_m^P}$ while $\{Q_1, ...Q_n\}$ not. Because the dimension of $\varepsilon_{L_i^P}$ in $\{P_1, ...., P_m, Q_1, ...Q_n\}$ is $m$ and $Dim(L(\mathcal{S}_p)) = m$, $\omega^\mathsf{T}(P_1, ...., P_m, Q_1, ...Q_n)$ contains $\varepsilon_{L_i^P}$, for any $\omega \neq 0$. According to the Darmois-Skitovitch Theorem, we have $\omega^\mathsf{T}(P_1, ...., P_m, Q_1, ...Q_n) \not\perp\!\!\!\perp (P_{m+1}, ...., P_{2m})^\mathsf{T}$. That is to say, $(\{P_{(m+1)+1}, ...P_{2m}\}, \{P_1, ...., P_m, Q_1, ...Q_n\})$ violates the GIN condition.

Therefore, $L(\mathcal{S}_p) \to L(\mathcal{S}_q)$. □

## A.7  Proof of Proposition 3

**Proposition 3.** *Let $\mathcal{S}_1$ and $\mathcal{S}_2$ be two clusters of a LiNGLaM and $Dim(L(\mathcal{S}_1)) = Dim(L(\mathcal{S}_2))$. If $\mathcal{S}_1$ and $\mathcal{S}_2$ are overlapping, $\mathcal{S}_1$ and $\mathcal{S}_2$ share the same set of latent variables.*

*Proof.* Because $\mathcal{S}_1$ and $\mathcal{S}_2$ are overlapping, with loss of generality, assume that the shared element of $\mathcal{S}_1$ and $\mathcal{S}_2$ is $X_k$. Furthermore, we have that $L(\mathcal{S}_1)$ and $L(\mathcal{S}_2)$ are both parents of $X_k$. Based on the

definition of causal cluster and that $Dim(L(\mathcal{S}_1)) = Dim(L(\mathcal{S}_2))$, we have $L(\mathcal{S}_1) = L(\mathcal{S}_2)$. That is to say, $\mathcal{S}_1$ and $\mathcal{S}_2$ share the same set of latent variables. □

## A.8 Proof of Lemma 1

**Lemma 1.** *Let $\mathcal{S}_r$ be a cluster and $\mathcal{S}_k, k \neq r$ be any other cluster of a LiNGLaM. Suppose that $\mathcal{S}_r$ contains $2Dim(L(\mathcal{S}_r))$ number of variables with $\mathcal{S}_r = \{R_1, R_2, ..., R_{2Dim(L(\mathcal{S}_r))}\}$ and that $\mathcal{S}_k$ contains $2Dim(L(\mathcal{S}_k))$ number of variables with $\mathcal{S}_k = \{K_1, K_2, ..., K_{2Dim(L(\mathcal{S}_k))}\}$. if $(\{R_{Dim(L(\mathcal{S}_r))+1}, ...R_{2Dim(L(\mathcal{S}_r))}\}, \{R_1, ...., R_{Dim(L(\mathcal{S}_r))}, K_1, ...K_{Dim(L(\mathcal{S}_k))}\})$ follows the GIN condition, then $L(\mathcal{S}_r)$ is a root latent variable set.*

*Proof.* (i) Assume that $L(\mathcal{S}_r)$ is a root latent variable set. Due to the linearity assumption, there is no latent confounder between $L(\mathcal{S}_r)$ and another latent variable set. Based on Theorem 4, we have that $(\{R_{|L(\mathcal{S}_r)|}, ...R_{2Dim(L(\mathcal{S}_r))}\}, \{R_1, ...., R_{Dim(L(\mathcal{S}_r))}, K_1, ...K_{Dim(L(\mathcal{S}_k))}\})$ follows the GIN condition.

(ii) Assume that $L(\mathcal{S}_r)$ is not a root latent variable set, that is , $L(\mathcal{S}_r)$ has at least one parent set. Let $L(\mathcal{S}_p)$ be the parent of $L(\mathcal{S}_r)$ and $\mathcal{S}_p = \{P_1, P_2, ..., P_{2Dim(L(\mathcal{S}_p))}\}$. Thus, every element in $\{P_1, P_2, ..., P_{2Dim(L(\mathcal{S}_p))}\}$ has the component $\varepsilon_{L(\mathcal{S}_p)}$. Based on the definition of the GIN condition, we easily obtain that there is no $\omega \neq 0$ such that $\omega^\mathsf{T}\mathbb{E}[\{P_1, ...., P_{Dim(L(\mathcal{S}_p))}, R_1, ...R_{Dim(L(\mathcal{S}_r))}\}), (\{P_{Dim(L(\mathcal{S}_p))+1}, ...R_{2Dim(L(\mathcal{S}_p))})^\mathsf{T}] = 0$ because the dimension of $\varepsilon_{L(\mathcal{S}_p)}$ in $\{R_1, ...., R_{Dim(L(\mathcal{S}_r))}, K_1, ...K_{Dim(L(\mathcal{S}_k))}\}$ equals $Dim(L(\mathcal{S}_r))$. That is to say, $\omega^\mathsf{T}(R_1, ...., R_{Dim(L(\mathcal{S}_r))}, K_1, ...K_{Dim(L(\mathcal{S}_k))})$ must have the component $\varepsilon_{L(\mathcal{S}_p)}$. Thus, $\omega^\mathsf{T}(R_1, ...., R_{Dim(L(\mathcal{S}_r))}, K_1, ...K_{Dim(L(\mathcal{S}_k))})$ is dependent on $(P_{Dim(L(\mathcal{S}_p))+1}, ...R_{2Dim(L(\mathcal{S}_p))})^\mathsf{T}$ based on the Darmois-Skitovitch Theorem. Therefore, $(\{P_{Dim(L(\mathcal{S}_p))+1}, ...P_{2Dim(L(\mathcal{S}_p))}\}, \{P_1, ...., P_{Dim(L(\mathcal{S}_p))}, R_1, ...R_{Dim(L(\mathcal{S}_r))}\})$ violates the GIN condition.

From (ii), the lemma is proven. Moreover, from (i) and (ii), we show that $(\{R_{|L(\mathcal{S}_r)|}, ...R_{2Dim(L(\mathcal{S}_r))}\}, \{R_1, ...., R_{Dim(L(\mathcal{S}_r))}, K_1, ...K_{Dim(L(\mathcal{S}_k))}\})$ follows the GIN condition, if and only if $L(\mathcal{S}_r)$ is a root latent variable set. □

## A.9 Proof of Proposition 4

**Proposition 4.** *Suppose that $\{\mathcal{S}_1, ...\mathcal{S}_i, ..., \mathcal{S}_n\}$ are all the clusters of the LiNGLaM. Denote $\mathbf{T} = \{L(\mathcal{S}_1), ...L(\mathcal{S}_i)\}$ and $\mathbf{T} = \{L(\mathcal{S}_{i+1}), ...L(\mathcal{S}_n)\}$, where all elements in $\mathbf{T}$ are causally earlier than those in $\mathbf{R}$. Let $\hat{\mathbf{Z}}$ contain the elements from the half children of each latent variable set in $\mathbf{T}$, and $\hat{\mathbf{Y}}$ contain the elements from the other half children of each latent variable set in $\mathbf{T}$. Furthermore, Let $L(\mathcal{S}_r)$ be a latent variable set of $\mathbf{R}$ and $\mathcal{S}_r = \{R_1, R_2, ..., R_{2Dim(L(\mathcal{S}_r))}\}$. If for any one of the remaining $\mathcal{S}_k \in \mathbf{R}, k \neq r$ and $\mathcal{S}_k = \{K_1, K_2, ..., K_{2Dim(L(\mathcal{S}_k))}\}$ such that $(\{R_{Dim(L(\mathcal{S}_r))+1}, ..., R_{2Dim(L(\mathcal{S}_r))}, \hat{\mathbf{Z}}\}, \{R_1, ...., R_{Dim(L(\mathcal{S}_r))}, K_1, ...K_{Dim(L(\mathcal{S}_k))}, \hat{\mathbf{Y}}\})$ follows GIN condition, then $L(\mathcal{S}_r)$ is a root latent variable set in $\mathbf{R}$.*

*Proof.* One may treat the causally earlier sets as a new group. Then one can easily prove this result according to Lemma 1. □

# B   More experimental results of Synthetic data

Here, we add more results to show the performance of our algorithm for random generated graphs and more variables. In details, we generated graphs randomly with different numbers of latent variables, where each latent variable only have three observed variables. We run our method and obtain the following results in Table 1.

# C   More details of Real-Word data

For comparisons, we give the hypothesized factors formulated in [1] in Table 2.

Table 1: Results with different numbers of variables and randomly generated graphs (with sample size=2000).

| Number of variables (latent variables) | Latent omission | Latent commission | Mismeasurements | Correct-ordering rate |
|---|---|---|---|---|
| 15(5) | 0.02(1) | 0.00(0) | 0.00(0) | 0.90 |
| 30(10) | 0.09(3) | 0.05(3) | 0.04(3) | 0.85 |
| 60(20) | 0.15(6) | 0.12(6) | 0.10(6) | 0.79 |

| Factors | Observed variables |
|---|---|
| *Role Conflict* | $RC_1, RC_2, WO_1, WO_2,$ |
| *Decision Making* | $DM_1, DM_2$ |
| *Classroom Climate* | $CC_1, CC_2, CC_3, CC_4$ |
| *Self-Esteem* | $SE_1, SE_2, SE_3$ |
| *Peer Support* | $PS_1, PS_2$ |
| *External Locus of Control* | $ELC_1, ELC_2, ELC_3, ELC_4, ELC_5$ |
| *Emotional Exhaustion* | $EE_1, EE_2, EE_3$ |
| *Denationalization* | $DP_1, DP_2, DP_3$ |
| *Personal Accomplishment* | $PA_1, PA_2, PA_3$ |

Table 2: The hypothesized factors in [1].

## Footnotes

[1]Note that we do not assume $\mathbf{E}_Z \perp\!\!\!\perp \mathbf{L}$.