[Reviews · NeurIPS 2020]

Review 1

Summary and Contributions: This is a paper that focuses on causal discovery especially focusing on the causal relation among latent variables. To this end, the authors propose a new latent variable model that is both linear and non gaussian. The main contribution is a recursive learning algorithm for locating latent variables and identifying their causal structure.

Strengths: Unmeasured confounders area challenge in causality, especially for causal discovery. The paper tackles this non-trivial task and studies the properties of the linear non-Gaussian model. Although the work relies on strong assumptions, it appears to be both novel and significant. The paper provides a thorough (theoretical) study of the GIN condition, which is closely related to the proposed algorithm. The analysis and the algorithm are sound and presented in a logical way.

Weaknesses: The complexity of algorithm is a cause for concern especially when the graph has a large number of nodes. This is because the algorithm involves the enumeration of all subsets. The algorithm is evaluated on four typical cases with few variables. It would have been more convincing with random generated graphs and more variables.

Correctness: Everything appears to be correct as far as I can tell from reading the paper. I've not carefully read the appendix.

Clarity: It would have been nice to have an example immediately after definition 1. Both writing and presentation in Section 3 should be substantially improved. There are several typos. After introducing Theorem 1 and Proposition 1, it would be good to further explain them. The exogenous set is a bit hard to follow in lines 166-170. An example would be very helpful. In the example in lines 170- 172, what are S1 and S2?

Relation to Prior Work: As far as I can tell, relation to prior work has been discussed well.

Reproducibility: Yes

Additional Feedback: Perhaps you should consider rephrasing "the incorrect ignorance of..." on line 26. In Definition 1 (A1), isn't it sufficient to say that no X should be a *parent* of any latent variable as opposed to saying no X should be an ancestor of any latent variable? In the text, explain the notation 2Dim() Either the proposition mentioned in line 113 should be placed in the main paper or the authors should simply not mention anything about it. *******Post Rebuttal********** Thanks for the clarifications and presenting the additional experiments. I am supporting the acceptance of this paper conditional on the authors addressing the concerns noted by all the reviewers, including a discussion on complexity and explicitly stating the limitations of their approach.


Review 2

Summary and Contributions: The paper introduces an algebraic condition and relates it to graphical structures that allow for the design of structure learning algorithms that discover relations among latent variables in the problem.

Strengths: - The paper builds on prior work in the field of structure learning in Linear Non-Gaussian models. - The authors put some effort into justifying these assumptions (at least through good references). - The experiments are quite thorough. - Overall, I think the theoretical contributions of this paper are interesting, and the experiments seem promising. There is a long history of learning the structure of graphical models and such works are always of interest to the machine learning community.

Weaknesses: - Some of the points I'll make here are more conceptual and would just like to hear from the authors what their thoughts are. (i) Gaussian vs non-Gaussian noise. Thanks for the reference to arguments relating to the ubiquity of non-Gaussian noise. However, there is another school of thought related to the Nonparanormal distribution that says everything can be transformed into something that looks Gaussian. In practice, either in applications or real-world analyses the authors have undertaken, what has been their experience in the usage of Gaussian vs non-Gaussian methods for structure learning. Is the first method they recommend a non-Gaussian method or one that relies on Gaussianity assumptions? (ii) Related to the above, the Silva et al 2006 paper that used Tetrad constraints did not really assume a specific family of distributions (just linearity) though much of the exposition uses multivariate Gaussian models for simplicity. In that sense, it is closer to algorithms like PC/FCI that are in theory nonparametric. So I do find it awkward to compare Triad constraints and the constraints in this paper as being strictly superior to Tetrad constraints. That is, multivariate Gaussian distributions are strictly excluded in these works. - There are certain sections of the paper that make statements without citation or more insights within the paper itself. Examples below: (i) "but practically, O-ICA is easy to get stuck in local optima, especially when the number of variables is relatively large". Is there a reference for this or are there experiments that demonstrate this? Isn't this the problem with any method trying to optimize a non-convex objective? Even PC is prone to propagating errors if an earlier conditional independence test returns an incorrect answer. I don't think there's anything in the current proposal that protects from running into bad local optima if for example step 1 of the procedure returns incorrect causal clusters. (ii) "They focus on estimating the causal relationships between observed variables rather than that between latent variables. However, in real-world scenarios, it may not be the case–there are also causal structure over latent variables" I don't understand the reasoning in these sentences. The second sentence seems incomplete -- what may not be the case? There is always structure among the latent variables (even if it is the empty graph) and algorithms like FCI return a structure over the observed variables. There are many problems in which investigators are not interested in the structure over latents and for these purposes algorithms like FCI are desirable. I understand the present algorithm occupies a different space of interest such as fields in which latent factor analysis is very common and it is interesting in its own right. However, the comparison to FCI and other algorithms like it in this particular context seems like a type error. - As a point of future work I'd be eager to see an algorithm that relaxes the purity assumption in some way. I've always found that assumption to be the least realistic.

Correctness: - The timeline for review this year has been quite compressed so I have not been able to check the math as thoroughly as I would like but it seemed fine to me.

Clarity: - Aside some of the points raised above the paper is written ok. - There are several typos throughout the paper that need fixing however. E.g. causla, LiNGMLaM

Relation to Prior Work: - I think the exact differences between the Triad constraints and the present work should be highlighted. - Other issues I had with discussion of previous work has already been mentioned above.

Reproducibility: Yes

Additional Feedback: Good work, look forward to more! :) ************* Feedback after author response/reviewer discussion ************* - I agree with Reviewer #4 that some of the assumptions need to be brought forward and highlighted more in the abstract/intro. Assumptions such as purity, even though they are very common in the structure learning literature that is specifically of this flavor, do need to be conveyed to folks who may be interested in applying the methodology in their own research. The more they are aware of what the underlying assumptions are the better. With all that said, it seems the authors also agree and will make such changes. - The authors also address reviewer concerns with additional experiments with randomly generated graphs. - The authors did not address the concern raised by Reviewer #1 regarding computational complexity of their algorithm. My take on this (and the authors should probably clarify if this is true) is that Algorithm 1 can stop before enumerating all possible subsets as the stopping criteria is until P is empty. I guess this is sort of similar to the PC/FCI algorithm which in theory could go up to all possible conditioning sets but if the graph is sparse enough or has some upper bound on the degree of each node, it can terminate much sooner. Same with GES, which in theory could end up going to a complete graph in the process of forward search if there's no assumptions on sparsity. I think this is an important point, that the authors should clarify in revisions of their paper. - Overall, I think the authors addressed reviewer feedback reasonably well and this inclines me to keep my score at a 7, advocating for acceptance.


Review 3

Summary and Contributions: In this paper, the authors propose a novel method to discover the causal relation between the confounders of the observed variables. They consider the Linear Non-Gaussian setting, in which the confounders are causally related. The authors provide solid theoretical guarantees for their method. The experimental results demonstrate the effectiveness of their approach.

Strengths: This paper is well-written and well-organized. The authors have an in-depth understanding of the related works and provide a detailed review. The paper tackles a very difficult problem and provide surprising results despite some strict assumptions. The theoretical guarantee for their method is solid. And the method is novel. In addition to the mentions by the authors, the thing that makes me excited is that I think the technique in this paper may be applicable in stock market, in which each individual stock is a covariate and we can cluster different stocks by their method.

Weaknesses: The assumptions A.4. seems a bit strict.

Correctness: The proofs seem to be correct and technically sound. But I did not check them throughly.

Clarity: Yes.

Relation to Prior Work: Yes.

Reproducibility: Yes

Additional Feedback: I wonder whether the proposed causal discovery algorithm is complete. If the algorithm cannot identify one causal relation, is it possible that the edge is identifiable but the method fails to do that? Besides, I think there might be a typo. In Algo. 1 Line 4, is it Dim(S) or Dim(P)? I might have some misunderstandings about this paper. So it is very likely that I will change my score if the other reviewers or authors provide some new points. After rebuttal: Thanks for the clarification. I am very happy to see it accepted.


Review 4

Summary and Contributions: The authors consider the problem of estimating causal graphs in the presence of latent variables. They consider specifically linear non-gaussian models and estimation of causal relationships among the latent variables. An algorithm for identifying latent variables and their structure based on a new criterion (GIN) is proposed. The algorithm consists of two stages, where first clusters of observed variables are identified and then an ordering is estimated. They demonstrate their algorithm’s performance on synthetic and real data.

Strengths: The paper considers an important problem of learning causal graphs in the presence of latent variables. Latent variables are often a hurdle for many causal inference methods. The authors provide empirical evaluation of their algorithm, which includes comparison with multiple prior methods. In addition, the algorithm is applied to real data and seems to give reasonable results.

Weaknesses: The authors should state their exact assumptions, especially the requirement that there are no edges between observed variables earlier in the paper (preferably in the abstract). These assumptions should also be motivated with real world examples and how reasonable they might be. In line 90, assumption A3 does not seem much milder than Tetrad since the difference is in only 1 observed variable. The authors can include additional prior work. The paper by Anandkumar et al. seems relevant since I believe they also consider the case of latent variables and learn causal relationships between them. A. Anandkumar, D. Hsu, A. Javanmard, and S. Kakade. Learning linear bayesian networks with latent variables. In International Conference on Machine Learning, pages 249–257, 2013. It seems that the algorithm additionally assumes that observed variables are a linear function of the latent variables. This could be additionally discussed since in real world settings this may not be the case. I thought that Algorithm 2 would return a causal graph over the causal clusters as opposed to an order. Please clarify this part. I think that the evaluation on synthetic data can be strengthened by including an additional experiment where DAGs are sampled randomly instead of fixing an order. Otherwise, perhaps the shown gains in performance might be only applicable to these specifically constructed causal graphs. Some claims on the empirical observations appear to not be exactly correct such as in line 291 the authors say GIN achieves lowest error on all cases, this makes it seem like the error of GIN is always the lowest, however, that’s not the case. For example the entry for Case 3, N=500, latent commission column, GIN has the highest error.

Correctness: Most of the claims appear to be correct, although I did not thoroughly check.

Clarity: The paper could be improved in terms of clarity. There are many typos. I point out some specific parts that need further clarification It would be great to get a preview of all the theorems and what do they generally show in the beginning of the paper as well as how they all tie together. Line 244: what do the boxes indicate? Line 126: typo, should be causal structure

Relation to Prior Work: The authors discuss and compare to prior work. However, some additional literature could be added for comparison (see above).

Reproducibility: Yes

Additional Feedback: ************* Feedback after author response/reviewer discussion ************* Thanks for doing additional experiments with random graphs and further clarifications. I think this paper still needs to be improved a little in terms of writing, so that would be great to see.

[Author Response · NeurIPS 2020]

We are grateful to the reviewers for the insightful comments and suggestions. Please see below for our response.

**To R1**: Q1. with randomly generated graphs and more variables.: Following your suggestion, we have generated graphs randomly with different numbers of latent variables. See Table 1 below for the results. Will include the results in the paper. Q2. "have an example after definition 1": Thanks for the suggestion. Will do it. Q3. "further explain Theorem 1 and Proposition 1.": Thanks. We will give more explanations about Theorem 1, as well as Prop. 1., in light of the example in Fig. 1. Specifically, Prop. 1 inspires a unified method to handle causal relations between latent variables and those between latent and observed variables; see the discussion in Section 6. Q4. "The example about exogenous set, S1 and S2 in lines 166-172.": In the first case, $S_1$ is an exogenous set relative to variable set $S_2$, where $S_1 = \{L_1\}$ and $S_2 = \{L_3, L_4\}$. In the second case, $S_1$ is not an exogenous set relative to variable set $S_2$, where $S_1 = \{L_2, L_3\}$ and $S_2 = \{L_3, L_4\}$. Will give more details. Q5. "In Definition 1 (A1), sufficient to say that no $\mathbf{X}$ should be a parent of any latent variable as opposed to saying no $\mathbf{X}$ should be an ancestor of any latent variable?": Yes, they are equivalent here. We followed the definition in [12] and will give this interpretation. Q6. "the notation $2Dim(\mathbf{L})$": $2Dim(\mathbf{L})$ means 2 times the dimension of $\mathbf{L}$. Will clarify it.

**To R2**: Q1. "everything can be transformed into something that looks Gaussian": Yes we agree; however, when such transforms are applied, the relationships between the transformed variables might not be linear. We think the Gaussian and non-Gaussian methods are complementary and have their own strengths. Because Gaussian methods only use the second order statistics, they have wider applicability. Non-Gaussian methods can provide more information of the structure, in light of the high order statistics. In practice, one may decide which method to apply first based on whether the data are linear and non-Gaussian (e.g., as seen from the scatter plots of the variables). Q2. "multivariate Gaussian distributions are strictly excluded in these works.": Yes. In the multivariate Gaussian case, one can apply traditional Tetrad-based methods, although there is no additional structural information informed by non-Gaussianity. Q3. "O-ICA is easy to get stuck in local optima...": We found that O-ICA is easy to get stuck in local optima, unless the underlying sources are very sparse. This was also reported in publications "Discovering unconfounded causal relationships using linear non-gaussian models" (by Entner, et al., 2011) and "ParceLiNGAM: A causal ordering method robust against latent confounders" (by Tashiro, et al., 2014). However, to avoid possible confusion, we will remove this statement. Q4. "what may not be the case?": Here, we mean that focusing on causal relationships between observed variables alone may not be enough, and the causal structure over latent variables might be very informative. For example, in some cases, the measured variables (such as questionnaire answers) may not loyally reflect the underlying variables of interest and the interesting causal process is over the latents. Q5. "the exact differences between the Triad constraints and the present work should be highlighted.": Thanks for your suggestion. The Triad condition can be seen as a restrictive, special case of the GIN condition, where $Dim(Y) = 2$ and $Dim(Z) = 1$. We will make it explicit.

**To R3**: Q1. "Algorithm is complete?": Yes, it is complete, as implied by Theorem 3 and Proposition 2 (for step 1) and Theorem 4 and Proposition 3 (for step 2). Will make it explicit. Q2. "is it Dim(S) or Dim(P)?": Thanks for pointing out the typo. It should be Dim(P). Has been corrected.

**To R4**: Q1. "State assumptions earlier...motivated with real world examples": The assumptions were explicitly given as A1-A4, in the definition of LiNGLaM. Following your suggestion, we will include the assumptions in Abstract and give illustrative real examples. Q2. "A3 does not seem much milder than Tetrad": Compared to Tetrad-based methods, our proposal involves less restrictive structural assumptions but produces stronger results. For instance, under the non-Gaussianity assumption, the graph in the Figure 1 can be recovered by the proposed method, but not by Tetrad-based methods. Q3. "The paper by Anandkumar et al.": Thanks for the suggested this interesting work. This paper makes use of non-Gaussainity of latent variables and was innovative; we will discuss its connection to and difference from our model. We are doing empirical comparisons with it. Q4. "Algorithm 2 returns a causal graph as opposed to an order.": Algorithm 2 indeed returns a causal order of the latent variables. Based on the order, one may directly obtain the causal structure by further estimating the linear coefficients and pruning redundant edges; please see the discussion in lines 333-337. Q5. "Additional experiment where DAGs are sampled randomly.": Thanks for the helpful suggestion. Table 1 below gives the results with randomly generated graphs. Q6. "error of GIN is always the lowest?": Many thanks for your careful observation. The *Mismeasurements* are higher in Case 3 when the sample size is small (N=500). We will update our claim and explain why this happens. Q7. "Line 244: the boxes indicate?": The boxes indicate the elements of the root variable set $\{L_1, L_2\}$. Will explain it in the paper.

Table 1: Results with different numbers of variables and randomly generated graphs (sample size=2000).

| Number of variables (latent variables) | Latent omission | Latent commission | Mismeasurements | Correct-ordering rate |
|---|---|---|---|---|
| 15(5) | 0.02(1) | 0.00(0) | 0.00(0) | 0.90 |
| 30(10) | 0.09(3) | 0.05(3) | 0.04(3) | 0.85 |
| 60(20) | 0.15(6) | 0.12(6) | 0.10(6) | 0.79 |

[Meta-Review · NeurIPS 2020]

The paper introduces a structure learning method via a relationship between an algebraic condition and structures in a graphical model. The reviewers felt this was a novel and interesting contribution to the structure learning literature, and the experimental evaluation provided by the authors was found to be quite thorough.